# MedAgentGym: A Scalable Agentic Training Environment for Code-Centric Reasoning in Biomedical Data Science

**Ran Xu**[1]*, **Yuchen Zhuang**[2]*, **Yishan Zhong**[2], **Yue Yu**[2], **Zifeng Wang**[3], **Xiangru Tang**[4],
**Hang Wu**[2], **May D. Wang**[2], **Peifeng Ruan**[5], **Donghan Yang**[5], **Tao Wang**[5],
**Guanghua Xiao**[5], **Xin Liu**[6], **Carl Yang**[1], **Yang Xie**[5]†, **Wenqi Shi**[5]†

Emory University[1] Georgia Institute of Technology[2] University of Illinois Urbana-Champaign[3]
Yale University[4] UT Southwestern Medical Center[5] University of Washington[6]

⬡ **MedAgentGym:** https://github.com/wshi83/MedAgentGym

🤗 **MedAgentGym:** https://huggingface.co/MedAgentGym

## Abstract

We introduce MedAgentGym, a scalable and interactive training environment designed to enhance coding-based biomedical reasoning capabilities in large language model (LLM) agents. MedAgentGym comprises $72,413$ task instances across $129$ categories derived from $12$ authentic real-world biomedical scenarios. Tasks are encapsulated within executable sandbox environments, each featuring detailed task specifications, interactive feedback mechanisms, verifiable ground truth annotations, and scalable training trajectory generation. Extensive benchmarking of 29 LLMs reveals substantial performance disparities in biomedical data science between commercial and open-source LLMs. Leveraging efficient multi-threaded and multi-turn trajectory sampling in MedAgentGym, Med-Copilot achieves performance gains of $+43.02\%$ and $+45.28\%$ from offline and online reinforcement learning, respectively, demonstrating MedAgentGym as an effective training ground while establishing itself as a cost-effective, privacy-preserving alternative competitive with proprietary LLMs (gpt-4o). By offering a unified execution environment with a comprehensive benchmark and accessible, extensible training resources, MedAgentGym delivers an integrated platform to develop LLM-based coding assistants for advanced biomedical data science.

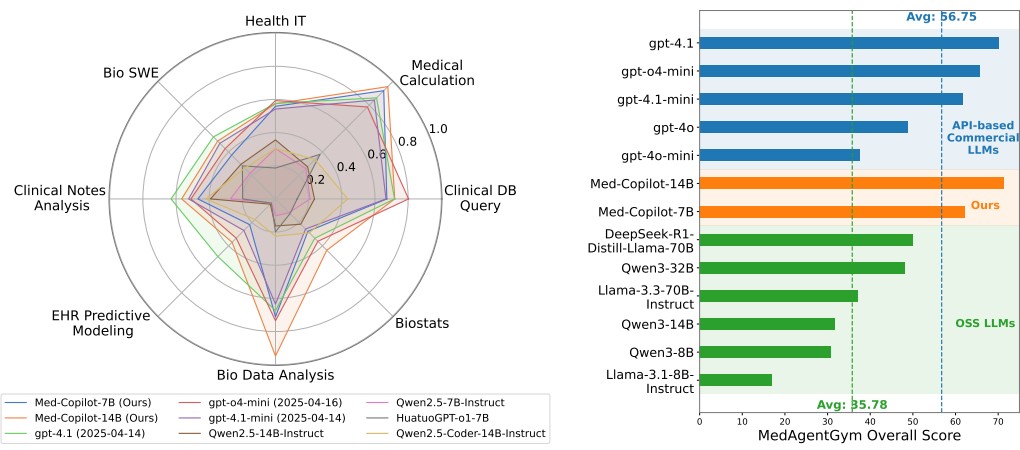

(a) Biomedical coding capabilities with MedAgentGym

(b) Overall score of MedAgentGym

Figure 1: Overview of (a) task-specific and (b) overall leaderboard evaluation in MedAgentGym. The results show the (a) performance variations across biomedical data science tasks and (b) large gaps between proprietary and open-source (OSS) LLMs, highlighting the need for continued development of privacy-preserving, affordable LLM agents, especially for complex code-based biomedical reasoning tasks such as biomedical software engineering and predictive modeling.

---

*Equal contribution. †Correspondence to: {Yang.Xie,Wenqi.Shi}@UTSouthwestern.edu.

# 1 INTRODUCTION

The exponential growth of healthcare data has fundamentally transformed modern biomedical research, intensifying the need for integration of advanced computational methods with medical domain expertise (Wornow et al., 2023b; Liu et al., 2025b). Biomedical researchers routinely face data science challenges that demand both medical data analysis knowledge and programming proficiency, such as querying large-scale databases, conducting statistical analyses, processing genomic sequences, and building predictive models from electronic health records (EHRs) (Nimmolrat et al., 2021; Lee et al., 2022; Wornow et al., 2023a). While recent advances in large language models (LLMs) have demonstrated significant capabilities in advanced reasoning (OpenAI, 2025b; Guo et al., 2025), including code generation (DeepMind, 2025) and scientific discovery (Swanson et al., 2024; Team et al., 2025; Yuan et al., 2025), it remains challenging to translate real-world biomedical data science requirements into executable computational solutions (Wang et al., 2024b; 2025d).

Developing effective biomedical coding agents poses unique challenges beyond knowledge-intensive medical reasoning (Wang et al., 2025b;c) and general-purpose code generation (Zheng et al., 2024; Jing et al., 2025). Within biomedical research and clinical practice, direct deployment of proprietary LLMs remains infeasible due to strict privacy requirements and prohibitive operational costs (Meskó & Topol, 2023; Shi et al., 2024a), whereas OSS LLMs exhibit substantial deficiencies in biomedical coding capabilities (Figure 1). Mitigating this performance disparity calls for addressing two infrastructure gaps: (1) *comprehensive, code-centric biomedical reasoning benchmarks* to diagnose agent limitations and support rigorous, reproducible evaluation; and (2) *specialized, interactive training environments* to develop the complex reasoning and robust coding capabilities required for real-world biomedical data science.

In this study, we introduce `MedAgentGym`, a scalable and agentic training environment designed to systematically enhance the coding-centric reasoning capabilities of LLM agents for biomedical data science workflows. Grounded in diverse real-world biomedical scenarios, `MedAgentGym` provides:

- **Comprehensive suite of code-centric biomedical reasoning tasks.** `MedAgentGym` encompasses 72,413 biomedical *coding-centric* instances across 129 categories grounded in 12 real-world biomedical scenarios[1]. We standardize a rich collection of biomedical data science tasks as executable problems with verifiable ground truth, spanning structured medical information retrieval, numerical clinical reasoning, bioinformatics, and machine learning (ML) modeling. Tasks incorporate diverse data modalities, including EHR tables, clinical notes, genomics, drugs, and biological sequences, which require medical domain-specific reasoning capabilities.

- **Scalable and interactive training infrastructure.** `MedAgentGym` provides an optimized, user-friendly environment to accelerate agent training. Each instance is encapsulated within *executable, isolated, and reproducible* Docker environments with pre-install dependencies, supporting multi-threading, parallel execution, and sequential sampling. `MedAgentGym` ensures efficient trajectory collection and facilitate large-scale automated evaluation compatible with diverse agent scaffolds.

- **Extensive benchmarking and effective agent training for biomedical data science.** Through an extensive benchmark of 29 proprietary and open-source LLMs, we identify critical deficiencies in biomedical data analysis and predictive modeling. `MedAgentGym` effectively strengthens agentic training: `Med-Copilot-7B` achieves gains of +43.02% and +45.28% through offline and online reinforcement learning (RL), respectively, and performs comparably to `gpt-4o` on both in- and out-of-distribution tasks. We publicly release `MedAgentGym` and `Med-Copilot`, together with high-quality training trajectories and the outcome verifier, to support reproducible benchmarking and continued development of LLM coding agents in biomedical data science.

# 2 RELATED WORKS

**Coding-Centric Reasoning in Biomedical Data Science.** Most existing medical benchmarks primarily evaluate LLMs on knowledge-intensive, narrative reasoning (Jin et al., 2019; Pal et al., 2022; Tsatsaronis et al., 2015). Although several efforts target isolated biomedical algorithmic tasks (Tang et al., 2024a; HAI@Stanford, 2025; Wang et al., 2024b) or simulate portions of clinical

---

[1]We emphasize that `MedAgentGym` mainly focuses on computational *code generation* for biomedical reasoning, rather than traditional medical coding systems (Soroush et al., 2024) such as ICD-9 or ICD-10.

Table 1: Summary of related biomedical reasoning and coding datasets with task details and execution environments. MedAgentGym is among the first publicly available training environments for LLM agents in biomedical data science, uniquely integrating *executable environments, interactive feedback, and task-isolated run-time facilities* for coding-based reasoning. "DB", "DA", "Bioinfo", and "ML" denote "database", "data analytics", "bioinformatics", and "machine learning", respectively.

| Datasets (↓) | Domain | | Task | | | | Environment & Facility | | | | Scale (#Instances) | | |
|---|---|---|---|---|---|---|---|---|---|---|---|---|---|
| | QA | Coding | DB | DA | Bioinfo | ML | Execution | Interaction | Isolation | Training | # Train | # Test | # Traj. |
| MedMCQA (Pal et al., 2022) | ✔ | ✗ | ✗ | ✗ | ✗ | ✗ | ✗ | ✗ | ✗ | ✗ | 3K | 4.18K | ✗ |
| MedQA (Jin et al., 2021) | ✔ | ✗ | ✗ | ✗ | ✗ | ✗ | ✗ | ✗ | ✗ | ✗ | 11.4K | 1.27K | ✗ |
| PubMedQA (Jin et al., 2019) | ✔ | ✗ | ✗ | ✗ | ✗ | ✗ | ✗ | ✗ | ✗ | ✗ | 450 | 500 | ✗ |
| BioASQ (Tsatsaronis et al., 2015) | ✔ | ✗ | ✗ | ✗ | ✗ | ✗ | ✗ | ✗ | ✗ | ✗ | 745 | 140 | ✗ |
| MedAgentsBench (Tang et al., 2025) | ✔ | ✗ | ✗ | ✗ | ✗ | ✗ | ✗ | ✗ | ✗ | ✗ | – | 862 | ✗ |
| MIRAGE (Xiong et al., 2024) | ✔ | ✗ | ✗ | ✗ | ✗ | ✗ | ✗ | ✗ | ✗ | ✗ | – | 7.66K | ✗ |
| HealthBench (Arora et al., 2025) | ✔ | ✗ | ✔ | ✗ | ✗ | ✗ | ✗ | ✗ | ✗ | ✗ | – | 5K | ✗ |
| EHRSQL (Lee et al., 2022) | ✗ | ✔ | ✔ | ✗ | ✗ | ✗ | ✗ | ✗ | ✗ | ✗ | 15.5K | 1.73K | ✗ |
| MedCalcBench (Khandekar et al., 2024) | ✗ | ✗ | ✗ | ✔ | ✗ | ✗ | ✗ | ✗ | ✗ | ✗ | 10.1K | 1.05K | ✗ |
| MedAgentBench (Jiang et al., 2025b) | ✗ | ✔ | ✗ | ✔ | ✗ | ✗ | ✔ | ✔ | ✗ | ✗ | – | 300 | ✗ |
| BioCoder (Tang et al., 2024a) | ✗ | ✔ | ✗ | ✔ | ✔ | ✗ | ✔ | ✗ | ✔ | ✗ | – | 1.24K | ✗ |
| BioDSBench (Wang et al., 2024b) | ✗ | ✔ | ✗ | ✔ | ✔ | ✗ | ✔ | ✔ | ✗ | ✗ | – | 128 | ✗ |
| EHRSHOT (Wornow et al., 2023a) | ✗ | ✔ | ✗ | ✗ | ✗ | ✔ | ✗ | ✗ | ✗ | ✗ | – | 15 | ✗ |
| **MedAgentGym (Ours)** | ✗ | ✔ | ✔ | ✔ | ✔ | ✔ | ✔ | ✔ | ✔ | ✔ | 59.2K | 13.2K | 6.7K |

workflows (Schmidgall et al., 2024; Li et al., 2024c;b), they do not capture a complete set of tasks in the full end-to-end lifecycle of biomedical data science, from data extraction (Lee et al., 2022; Ryu et al., 2024) to model development (Wornow et al., 2023a; Wang et al., 2020b). Complementing these benchmarks, MedAgentGym emphasizes computation- and coding-intensive tasks that require LLM agents to retrieve, transform, analyze, and compute biomedical data while generating and executing code with pre-installed biomedical libraries and dependencies to produce verifiable solutions.

**Scalable and Interactive Training Environment for Biomedical Coding Agents.** Agentic RL (Guo et al., 2025; Schulman et al., 2017; Shao et al., 2024b) shifts LLM post-training from passive sequence generation to autonomous agents operating in complex, dynamic settings, including medical reasoning (Xia et al., 2025; Jiang et al., 2025a; Chen et al., 2024; Lan et al., 2025; Wu et al., 2025a; Wang et al., 2025a). Within such a framework, agents interact iteratively with their environment, receiving observations and executing actions, while the environment returns reward signals and state updates (Wang et al., 2025e; Chezelles et al., 2024; Shao et al., 2024a; Nathani et al., 2025). However, most biomedical reasoning and data science benchmarks (Table 1) are single-pass evaluations without executable environments or agent-level interaction signals (Zhu et al., 2025; Arora et al., 2025; Wu et al., 2025b). In contrast, MedAgentGym uniquely provides an executable and interactive biomedical coding environment covering comprehensive range of tasks. It also supports efficient multi-turn trajectory sampling through multi-threaded rollouts, thus enabling scalable and systematic improvement via agentic fine-tuning beyond prompting (Shi et al., 2024b; Huang et al., 2025a).

# 3 MEDAGENTGYM: A SCALABLE AND INTERACTIVE LLM AGENT TRAINING ENVIRONMENT FOR CODE-CENTRIC BIOMEDICAL REASONING

## 3.1 PROBLEM FORMULATION

We formulate coding-based reasoning as a structured problem-solving task: given a problem description $x \in \mathcal{X}$, the goal is to generate a code snippet $c \in \mathcal{C}$ that produces an output $y \in \mathcal{Y}$. Each instance $(x, y)$ is paired with a ground truth output $y^*$, and the correctness is verified using $\mathcal{E} : \mathcal{C} \times \mathcal{Y} \rightarrow \{0, 1\}$, where $\mathcal{E} = \mathbb{I}(y = y^*)$. Existing biomedical reasoning datasets typically provide only question-answer pairs $(x, y^*)$ without code solutions $c$ or only include a single predefined code solution per task. To address this, MedAgentGym enables scalable generation and sampling of multiple coding trajectories $c^{(0)}, c^{(1)}, \cdots, c^{(k)}$ with corresponding executions $y^{(0)}, y^{(1)}, \cdots, y^{(k)}$ through parallel execution of LLM agents. Each trajectory is either single-turn or multi-turn, depending on task complexity and user requirements. Crucially, MedAgentGym captures both *positive* trajectories $\{c^{(i)} | y^{(i)} = y^*\}$ that succeed and *negative* trajectories $\{c^{(i)} | y^{(i)} \neq y^*\}$ including error messages as learning signals.

## 3.2 DATA CONSTRUCTION: FROM INDIVIDUAL DATASETS TO UNIFIED BENCHMARK

**Task and Data Identification.** MedAgentGym focuses on verifiable biomedical data science tasks that benefit from code-based solutions (*i.e.*, code-centric biomedical reasoning). *Clinically*, we prioritize

Table 2: Dataset statistics for `MedAgentGym` and its lightweight subset for leaderboard evaluation. *For open-ended tasks without explicit ground truth (*e.g.*, ML coding in EHRSHOT and MIMIC-Extract), we follow standard RL settings by using the same dataset for training and evaluation.

| Dataset | Data Sources | | | | Task Instances (all) | | | | Tasks (leader-board) | | |
|---|---|---|---|---|---|---|---|---|---|---|---|
| | Type | #Patients | #Table | #Elements | Category | #Train | #Test | #Total | #Train | #Test | #Total |
| *Training and Internal Validation (In-Distribution)* | | | | | | | | | | | |
| MIMIC-III (Johnson et al., 2016) | Tabular | <1K | 17 | 1.4M | 9 | 9,318 | 1,122 | 10,440 | 552 | 581 | 1,133 |
| eICU (Pollard et al., 2018) | Tabular | <1K | 10 | 1.5M | 9 | 6,213 | 611 | 6,824 | 559 | 610 | 1,169 |
| TREQS (Wang et al., 2020a) | Tabular | 100 | 5 | 2.5M | 4 | 8,988 | 996 | 9,984 | 897 | 995 | 1,892 |
| MedCalcBench (Khandekar et al., 2024) | Text | 1K | – | – | 55 | 10,053 | 1,047 | 11,100 | 1,005 | 1,046 | 2,051 |
| MedAgentBench (Jiang et al., 2025b) | Tabular | 100 | – | 700K | 10 | 433 | 109 | 542 | 239 | 59 | 298 |
| BioCoder (Tang et al., 2024a) | Text | – | – | – | 8 | 981 | 157 | 1,138 | 981 | 156 | 1,137 |
| EHRSHOT (Wornow et al., 2023a) | Tabular | 63K | 31 | 1.2M | 15 | 15 | 15 | 15* | 15 | 15 | 15* |
| BioDSBench (Wang et al., 2024b) | Text | – | – | – | 12 | 50 | 49 | 99 | 50 | 49 | 99 |
| **MedAgentGym (Internal)** | – | 65K | 63 | 7.3M | 113 | 36,036 | 4,106 | 40,142 | 4,283 | 3,511 | 7,794 |
| *External Validation (Out-of-Distribution)+ only the test set for external evaluation; training data remains accessible* | | | | | | | | | | | |
| EHR-SeqSQL (Ryu et al., 2024) | Tabular | <1K | 17 | 1.4M | 4 | 18,950 | 7,913 | 26,863 | 1,000 | 500 | 1,500 |
| EHRCon (Kwon et al., 2024) | Tab&Text | 46K | 13 | – | 3 | 3,229 | 976 | 4,205 | 1,000 | 500 | 1,500 |
| MIMIC-Extract (Wang et al., 2020b) | Tabular | 35K | 4 | 35K | 3 | 3 | 3 | 3* | 3 | 3 | 3* |
| N-PowerAI (Ruan et al., 2025) | Text | – | – | – | 6 | 960 | 240 | 1200 | 960 | 240 | 1200 |
| **MedAgentGym (External)** | – | 82K | 34 | 1.4M | 16 | 23,142 | 9,132 | 32,271 | 2,963 | 1,243 | 4,203 |
| *Overall* | | | | | | | | | | | |
| **MedAgentGym** | – | 146K | 80 | 7.4M | 129 | 59,175 | 13,238 | 72,413 | 7,243 | 4,754 | 11,997 |

tasks originating from real-world healthcare settings and validated by a multidisciplinary panel of healthcare experts. For example, `MedAgentGym` involves MIMIC-III and eICU in EHRSQL (Lee et al., 2022) collected from 222 hospital staff members and annotated by human programmers. *Computationally*, we integrate diverse coding tasks, ranging from *structured medical information retrieval* to *open-ended biomedical research*, ensuring comprehensive coverage and task diversity.

**Verifiable Instances Preparation.** To standardize tasks across various sources, each instance in `MedAgentGym` is structured with: (1) a problem description, (2) verifiable ground-truth outputs, and (3) optional data resources (*e.g.*, EHRs). Additionally, standardized system and user prompts are designed to initiate the problem-solving process (see appendix G). `MedAgentGym` is highly flexible, easily accommodating new tasks that include clear descriptions and verifiable ground-truth outputs. For coding-centric tasks that provide only reference code implementations (*e.g.*, BioCoder (Tang et al., 2024a)), we validate task correctness based on the execution output of these reference solutions, generating definitive output signatures. This transformation is necessary because multiple valid code implementations may yield identical execution results, making the execution outcome–rather than the code itself–a more reliable and consistent verification signal. For tasks involving additional data resources (*e.g.*, EHRSQL (Lee et al., 2022)), we include metadata on data access and sources. Detailed task overview and task-specific preparation are documented in appendix C.

**Data Statistics.** `MedAgentGym` is a unified training environment built upon a large-scale, high-quality dataset comprising approximately 72,000 task instances across 129 categories from 12 real-world biomedical scenarios. Notably, with `MedAgentGym`, we collect large-scale agent trajectories to support coding agent development (section 5). To ensure reproducible and robust evaluation, we define clear train/test splits, separate internal and external validation sets, and perform $n$-gram ($n = 10$) string match to eliminate the data contamination issue. Table 2 provides statistics for `MedAgentGym`. To accommodate diverse research needs, we offer two versions of `MedAgentGym`: (1) a comprehensive, full-scale dataset for extensive exploration and detailed analysis, and (2) a balanced, lightweight subset for efficient leaderboard training and evaluation.

### 3.3 CODING ENVIRONMENT: FROM STATIC BENCHMARK TO INTERACTIVE INTERFACE

**Isolated and Executable Sandbox Environment.** To ensure robust and reproducible coding-based biomedical reasoning, `MedAgentGym` provides isolated executable coding environments (*i.e.*, sandbox) through Docker containers tailored to each task (Figure 2). These containers come pre-installed with all required dependencies, including specialized biomedical packages (*e.g.*, AlignIO in BioCoder (Tang et al., 2024a)), facilitating reliable task execution. To address critical data safety concerns, each Docker environment guarantees: (1) *environmental integrity*, where isolation prevents contamination or data corruption potentially caused by LLM-generated code, preserving both the computational environment and the underlying data systems (Yang et al., 2024b); (2)

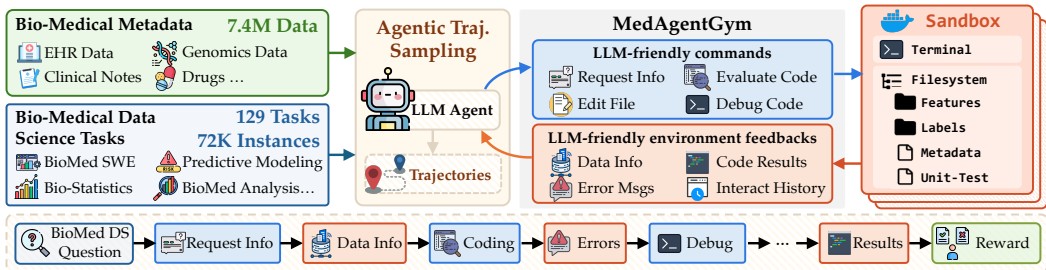

Figure 2: Overview of `MedAgentGym`. `MedAgentGym` contains a comprehensive suite of coding-centric biomedical data science tasks with an interactive execution environment for LLM agents.

*medical data security*, where secure containerization enforces compliance with medical data usage policies, safeguarding sensitive patient information. Additionally, `MedAgentGym` supports extensive flexibility for integrating new tasks, where users can define customized Docker environments through configuration files. If certain packages are not initially available, a terminal tool allows LLM agents to dynamically install the required dependencies within their isolated environments.

**Interactive Feedback.** `MedAgentGym` incorporates interactive feedback mechanisms, effectively bridging LLMs with coding interpreters: (1) *robust parsing:* To begin, the output generated by LLMs is formatted in structured JSON, facilitating straightforward parsing and code execution. In cases of execution errors, iterative JSON regeneration is employed to maximize successful code execution rates. (2) *debugging and error grounding:* Compile-time and runtime error messages are systematically translated into a unified natural language format, making them more accessible to LLMs and significantly improving debugging efficiency and interpretability.

**Efficient Trajectory Collection.** Each task in `MedAgentGym` is packaged in a reproducible Docker image with built-in support for *multi-threading*, *parallel execution*, and *sequential sampling*. Specifically, we integrate two widely used multi-threading backend engines, `Ray`[2] and `Joblib`[3], to accelerate trajectory sampling. This infrastructure ensures efficient and scalable trajectory collection, supporting both extensive experimentation and systematic evaluation across multiple scenarios.

**Plug-and-Play.** A key strength of `MedAgentGym` lies in its flexible and modular architecture, which readily supports the integration of new biomedical coding tasks. This inherent extensibility enables `MedAgentGym` to continually adapt to evolving advancements in biomedical sciences and artificial intelligence methodologies. Additionally, its trajectory sampling approach allows the straightforward transformation of traditional, non-executable biomedical reasoning tasks into coding-based scenarios with verifiable outputs, significantly broadening the scope and complexity of tasks that can be systematically evaluated. Moreover, users can define custom Docker environments through configuration files, and, if specific software packages are initially absent, a built-in terminal tool facilitates dynamic installation within each isolated execution environment, further improving `MedAgentGym` in runtime adaptability and user-friendliness.

## 4 EVALUATING LLMS AS MEDICAL CODING AGENTS WITH MEDAGENTGYM

### 4.1 EXPERIMENTS SETUP

**Agent Scaffolds.** Following CodeAct (Wang et al., 2024a), we establish a default agent scaffold for systematically evaluating coding-based biomedical reasoning. Interactions within `MedAgentGym` are modeled as a Partially Observable Markov Decision Process (POMDP), focusing on sampled biomedical data science tasks $p \in \mathcal{P}$. At each timestep $t$, the agent observes $o_t \in \mathcal{O}$ and samples an action $a_{t+1} \in \mathcal{A}$ from the current policy $\pi_t$ based on interaction history. We define four primary action types: (a) `request_info`: retrieve relevant data from sources such as EHRs; (b) `terminal`: manage dependencies or local files within isolated Docker environments. (c) `code_execution`: execute code generated by LLMs through an integrated interpreter; and (d) `debugging`: translate code execution errors into natural language explanations enriched with detailed error information for LLM comprehension.

---

[2]https://github.com/ray-project/ray
[3]https://joblib.readthedocs.io/en/stable/

Table 3: Test set results (zero-shot) of LLMs on MedAgentGym. **Bold** indicates the best result at each scale. ‡ and ∨ denote coding LLMs and medical reasoning LLMs, respectively.

| Datasets (→) Baselines (↓) / Metrics (→) | MIMIC. SR | eICU SR | TREQS SR | MedCalc. SR | MedAgent. SR | BioCoder SR | BioDS. SR | EHRSHOT Acc | Avg. Score |
|---|---|---|---|---|---|---|---|---|---|
| *API-based Proprietary LLMs†: We only consider Microsoft Azure OpenAI API services due to credentialed health data use agreement.* | | | | | | | | | |
| gpt-4o-mini (2024-07-28) (Hurst et al., 2024) | 35.97 | 16.57 | 38.39 | 73.11 | 40.38 | 30.12 | 57.35 | 7.84 | 37.47 |
| gpt-4o (2024-08-06) (Hurst et al., 2024) | 43.04 | 43.44 | 53.47 | 73.97 | 54.23 | 30.12 | 58.16 | 33.53 | 48.75 |
| gpt-4.1-mini (2025-04-14) (OpenAI, 2025a) | 62.79 | 63.44 | 69.75 | 84.36 | 54.23 | 47.46 | 63.47 | 48.28 | 61.72 |
| gpt-4.1 (2025-04-14) (OpenAI, 2025a) | 69.36 | 64.75 | 74.97 | **86.23** | 57.63 | **52.95** | 67.35 | **87.93** | **70.15** |
| gpt-o4-mini (2025-04-16) (OpenAI, 2025b) | **76.45** | **70.16** | 74.47 | 78.45 | **59.32** | 42.94 | **73.47** | 50.07 | 65.67 |
| ‡codex-mini (2025-05-16) (Chen et al., 2021) | 67.30 | 64.75 | 74.57 | 82.49 | 58.76 | 48.78 | 67.64 | 58.76 | 65.38 |
| *OSS (Base Size): < 10B parameters* | | | | | | | | | |
| Qwen3-1.7B (Qwen, 2025a) | 20.12 | 10.62 | 15.08 | 46.24 | 16.95 | 15.38 | 6.12 | 1.87 | 16.55 |
| Qwen3-4B (Qwen, 2025a) | 27.23 | 30.77 | 28.85 | 52.80 | 15.25 | 19.16 | 20.41 | 23.85 | 27.29 |
| gemma-3-4b-it (Gemma, 2025) | 27.36 | 29.10 | 24.52 | 42.49 | 18.64 | 17.95 | 8.16 | 4.37 | 21.57 |
| ∨medgemma-4b-it (Google, 2025) | 15.51 | 13.11 | 14.85 | 41.89 | 17.62 | 26.74 | 17.82 | 1.33 | 18.61 |
| Qwen2.5-7B-Instruct (Yang et al., 2024a) | 13.08 | 15.57 | 12.76 | 25.91 | 30.36 | 21.79 | 10.20 | 5.42 | 17.43 |
| Llama-3.1-8B-Instruct (Dubey et al., 2024) | 16.67 | 25.00 | 19.17 | 27.53 | 16.95 | 18.59 | 9.19 | 2.36 | 16.97 |
| ‡Qwen2.5-Coder-7B-Instruct (Hui et al., 2024) | 9.12 | 10.66 | 15.63 | 24.62 | 18.75 | 10.60 | 17.24 | 10.55 | 14.65 |
| ∨HuatuoGPT-o1-7B (Chen et al., 2024) | 4.99 | 7.04 | 7.04 | 38.05 | 18.64 | 28.21 | 19.88 | 5.03 | 16.11 |
| ∨m1-7B-23K (Huang et al., 2025b) | 6.88 | 9.56 | 7.04 | 28.24 | 9.32 | 20.26 | 14.71 | 0.00 | 12.00 |
| Qwen3-8B (Qwen, 2025a) | 29.08 | 34.53 | 37.37 | **54.59** | 20.34 | 20.51 | 24.49 | 25.71 | 30.83 |
| Ministral-8B-Instruct-2410 (Ministral, 2025) | 16.70 | 14.92 | 25.39 | 49.81 | 22.03 | 23.72 | 12.24 | 7.79 | 22.27 |
| ∨MedReason-8B (Wu et al., 2025a) | 9.12 | 9.51 | 9.15 | 43.31 | 21.46 | **31.42** | 17.42 | 3.88 | 18.16 |
| ‡Seed-Coder-8B-Reasoning (Seed et al., 2025) | **42.51** | **45.74** | **39.50** | 35.18 | 28.81 | 23.72 | 20.41 | 22.89 | **32.35** |
| *OSS (Large Size): 10 - 30B parameters* | | | | | | | | | |
| Qwen3-14B (Qwen, 2025a) | 31.50 | 31.97 | 30.05 | **61.38** | 22.03 | 22.60 | **26.53** | 26.77 | 31.60 |
| Qwen2.5-14B-Instruct (Yang et al., 2024a) | 17.21 | 14.07 | 16.43 | 27.40 | **35.59** | **29.49** | 16.33 | 4.45 | 20.12 |
| DeepSeek-R1-Distill-Qwen-14B (Guo et al., 2025) | 35.12 | 38.52 | 32.96 | 48.09 | 32.20 | 21.29 | 24.49 | 11.39 | 30.51 |
| ‡Qwen2.5-Coder-14B-Instruct (Hui et al., 2024) | **41.82** | **44.26** | **35.78** | 33.75 | 30.42 | 26.28 | 22.45 | **28.37** | **32.89** |
| ∨Baichuan-M1-14B-Instruct (Wang et al., 2025a) | 4.50 | 12.19 | 7.36 | 1.82 | 21.46 | 16.34 | 17.42 | 0.00 | 10.14 |
| *OSS (XL Size): > 30B parameters* | | | | | | | | | |
| Qwen3-32B (Qwen, 2025a) | 52.48 | 60.95 | 53.82 | 63.82 | 45.93 | 32.67 | 28.57 | 47.29 | 48.19 |
| Qwen2.5-32B-Instruct (Yang et al., 2024a) | 54.56 | 45.41 | 62.81 | 69.96 | 40.67 | 27.45 | 22.45 | 18.13 | 42.68 |
| QwQ-32B (Qwen, 2025b) | 62.31 | 56.72 | **66.15** | 67.69 | **47.46** | **42.31** | 14.29 | **55.05** | **51.50** |
| DeepSeek-R1-Distill-Qwen-32B (Guo et al., 2025) | 62.18 | 58.36 | 65.82 | 60.14 | 43.56 | 28.66 | 26.53 | 31.17 | 47.05 |
| ∨ Baichuan-M2-32B (Dou et al., 2025) | 20.83 | 23.61 | 24.92 | 30.02 | 25.42 | 25.00 | 20.41 | 12.94 | 22.89 |
| Llama-3.3-70B-Instruct (Dubey et al., 2024) | 39.93 | 25.08 | 24.92 | **84.99** | 39.40 | 27.55 | 24.49 | 29.93 | 37.04 |
| DeepSeek-R1-Distill-Llama-70B (Guo et al., 2025) | **64.59** | **64.92** | 56.98 | 76.96 | 28.81 | 32.05 | **42.86** | 33.42 | 50.07 |
| ∨ HuatuoGPT-o1-72B (Qwen2.5-72B) (Chen et al., 2024) | 27.19 | 29.84 | 29.65 | 52.01 | 28.81 | 31.41 | 26.53 | 16.87 | 30.29 |

**Tasks and Datasets.** Building upon MedAgentGym, we train and evaluate Med-Copilot on 7,794 *coding-based biomedical reasoning* tasks across 8 datasets: (1) MIMIC-III (Johnson et al., 2016) and (2) eICU (Pollard et al., 2018) from EHRSQL (Lee et al., 2022), (3) TREQS (Wang et al., 2020a), (4) MedCalcBench (Khandekar et al., 2024), (5) MedAgentBench (Jiang et al., 2025b), (6) BioCoder (Tang et al., 2024a), (7) EHRSHOT (Wornow et al., 2023a), and (8) BioDSBench (Wang et al., 2024b). Moreover, we conduct experiments for *out-of-distribution* evaluation on 4,203 tasks from the following 4 datasets: (9) EHR-SeqSQL (Ryu et al., 2024), (10) EHRCon (Kwon et al., 2024), (11) MIMIC-Extract (Wang et al., 2020b), and (12) N-PowerAI (Ruan et al., 2025). Note that we do not consider knowledge-intensive medical question-answering tasks (Jin et al., 2019; Pal et al., 2022; Jin et al., 2021), as they are orthogonal to coding-aided reasoning. We include detailed task and dataset information in appendix C.

**Baselines.** We extensively benchmark the following state-of-the-art LLMs on MedAgentGym: (i) *API-based proprietary LLMs*, including gpt-4o-mini (Hurst et al., 2024), gpt-4o (Hurst et al., 2024), gpt-4.1-mini (OpenAI, 2025a), gpt-4.1 (OpenAI, 2025a), gpt-o4-mini (OpenAI, 2025b), and codex-mini (Chen et al., 2021); (ii) *OSS LLMs*, including gemma-3 (Gemma, 2025), Qwen3 (Qwen, 2025a), Qwen2.5 (Yang et al., 2024a), Llama-3 (Dubey et al., 2024), Ministral (Ministral, 2025), and DeepSeek-R1 (Guo et al., 2025); (iii) *coding LLMs*, including codex-mini (Chen et al., 2021), Qwen2.5-Coder-7B-Instruct and -14B-Instruct (Hui et al., 2024), and Seed-Coder-8B-Reasoning (Seed et al., 2025); and (iv) *medical reasoning LLMs* or medical domain-specific LLMs, including medgemma-4b-it (gemma-3-4b-pt) (Google, 2025), HuatuoGPT-o1-7B (Qwen2.5-7B-Instruct) and HuatuoGPT-o1-72B (Qwen2.5-72B) (Chen et al., 2024), m1-7B-23K (Qwen2.5-7B-Instruct) (Huang et al., 2025b), MedReason-8B (Llama-3.1-8B-Instruct) (Wu et al., 2025a), Baichuan-M1-14B-Instruct (Wang et al., 2025a), and Baichuan-M2-32B (Dou et al., 2025). Additional model details are available in appendix D.

**Evaluation Metrics.** We adopt *success rate (SR)* as the primary evaluation metric. For *database, data science, and bioinformatics* tasks with explicit ground truths, we compare LLM-generated code execution outputs with reference solutions using exact match. For open-ended *ML* tasks in clinical decision support, we measure performance using *accuracy (Acc)* across test cases. See appendix E for implementation details and F.1 for additional evaluation on code quality and efficiency.

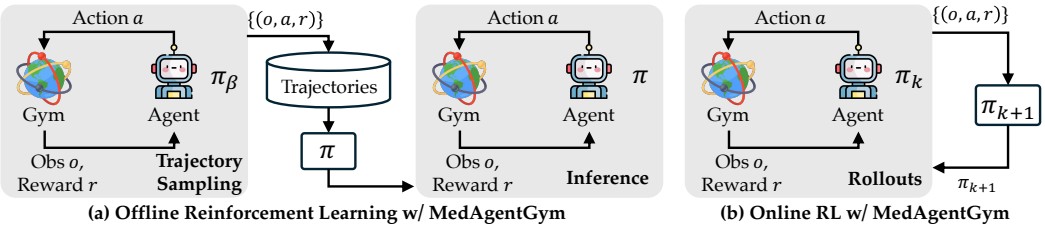

Figure 3: Comparison of (a) offline and (b) online RL paradigms within `MedAgentGym`.

## 4.2 RESULTS: BENCHMARKING LLMS AND REASONING MODELS WITH MEDAGENTGYM

Table 3 benchmarks the state-of-the-art LLMs on `MedAgentGym`. We summarize key observations from our zero-shot leaderboard evaluation as follows: ⋄ **Significant Performance Gap Between Commercial API-based and OSS LLMs.** This evident performance gap highlights the *critical need for continued development* of lightweight OSS LLMs that match commercial performance while addressing real-world privacy and cost constraints. ⋄ **Task-Specific Performance Variations between Structured and Open-ended Medical Tasks.** LLMs consistently perform better on structured tasks (*e.g.*, database queries, medical calculations) compared to open-ended tasks requiring advanced coding and reasoning (*e.g.*, data analysis, ML prediction). ⋄ **Suboptimal Outcomes in Dedicated Coding and Medical Domain-Specific LLMs.** Both coding and medical reasoning LLMs deliver limited improvement or even decline over base models, revealing that *coding-based biomedical reasoning represents a unique capability* not adequately captured by specialization in either coding or medical reasoning. Surprisingly, medical reasoning models (regardless of model sizes) consistently underperform relative to their base models except for knowledge-intensive tasks (*e.g.*, `MedCalcBench`, `BioCoder`), showing that fine-tuning in medical QA may reduce generalization and instruction-following ability. These findings highlight the need to jointly improve coding skills and medical reasoning, rather than treating them as separate objectives.

## 5 TRAINING LLM AGENTS FOR CODE-CENTRIC BIOMEDICAL REASONING

In this section, we leverage `MedAgentGym` to systematically enhance lightweight OSS LLMs as proficient coding agents (`Med-Copilot`) for biomedical reasoning. We first explore a two-stage agentic fine-tuning framework (section 5.1), followed by a detailed analysis of model scaling behaviors (section 5.2). We then introduce self-improvement to further boost agent performance (section 5.3) and conduct additional analysis on model generalization, ablation, and error patterns (section 5.4).

### 5.1 RL FINE-TUNING WITH TRAJECTORY SAMPLING

**Training Setup.** We select `Qwen-2.5-Instruct-7B` and `-14B` (Yang et al., 2024a) as our backbones. To enable effective evaluation within `MedAgentGym`, we utilize a consistent CodeAct-style scaffold, allowing LLM agents to iteratively reason and refine biomedical code through interactive environment feedback. Detailed training setups, including hyperparameters, are provided in appendix E.

**Trajectory Sampling.** `MedAgentGym` facilitates efficient parallel trajectory sampling using `ray` and `joblib` backends. Specifically, we roll out (1) 2,137 successful trajectories using `gpt-4.1-mini` with a temperature of 0 to warm up the fine-tuning for smaller OSS models. Each successful trajectory contains 9.25 turns between the LLM and the code interpreter on average. In addition to 2,137 positive trajectories for supervised fine-tuning (SFT), we prepare additional trajectory pairs for RL such as direct preference optimization (DPO), including (2) 1,646 off-policy preference pairs sampled from `gpt-4.1-mini`, and (3) 2,939 on-policy preference pairs. For both types, we use the initial prompt interactions as shared context and contrast successful final codes against intermediate erroneous attempts. In addition, we also performed a quantitative analysis on 250+ trajectories (randomly sampled over 10% of our trajectory collection) and confirmed that the vast majority of successful solutions followed a logically sound path, with cases of 'correct answer from flawed code' being exceptionally rare (<1%). We release all 6K trajectories above to accelerate coding agent development. See appendix C.6 for detailed trajectories composition.

**Two-Stage Fine-Tuning.** We benchmark two policy improvement methods: (1) SFT directly mimics high-reward trajectories consisting exclusively of successful outcomes, whereas (2) offline or online

Table 4: Med-Copilot performance on MedAgentGym finetuned with sampled trajectories.

| Datasets (→) Base (↓) / Metrics (→) | MIMIC-III SR | eICU SR | TREQS SR | MedCalc. SR | MedAgent. SR | BioCoder SR | BioDS. SR | EHRSHOT Acc | Avg. SR | Δ Score |
|---|---|---|---|---|---|---|---|---|---|---|
| Qwen2.5-7B-Instruct | 13.08 | 15.57 | 12.76 | 25.91 | 30.36 | 21.79 | 10.20 | 5.42 | 16.89 | – |
| +SFT | 57.83 | 61.48 | 72.66 | 89.06 | 50.85 | 28.33 | 55.10 | 15.62 | 53.87 | (+36.98) |
| +DPO | 64.13 | 66.91 | 72.02 | 90.06 | 52.54 | 34.62 | 69.39 | 29.55 | 59.90 | (+43.02) |
| +PPO | 66.10 | 67.25 | **73.88** | 74.52 | 51.33 | 32.71 | 65.47 | 32.40 | 57.96 | (+41.07) |
| +GRPO | **68.21** | **68.73** | 70.50 | **92.33** | **55.87** | **37.40** | **71.11** | 33.18 | **62.17** | (+45.28) |
| Qwen2.5-14B-Instrust | 17.21 | 14.07 | 16.43 | 27.40 | 35.59 | 29.49 | 16.33 | 4.45 | 20.12 | – |
| +SFT | 61.45 | 62.46 | 76.38 | 94.36 | 52.54 | 39.80 | 89.80 | 34.58 | 63.92 | (+43.80) |
| +DPO | 64.54 | 63.52 | 76.08 | 92.45 | 54.32 | 43.56 | 92.96 | 43.56 | 66.37 | (+46.25) |
| +PPO | 67.55 | 68.53 | **78.32** | 94.86 | 53.22 | 45.88 | 91.33 | 56.79 | 69.56 | (+49.44) |
| +GRPO | **68.78** | **69.34** | 76.84 | **95.81** | **57.41** | **49.32** | **94.78** | **59.05** | **71.42** | (+51.30) |

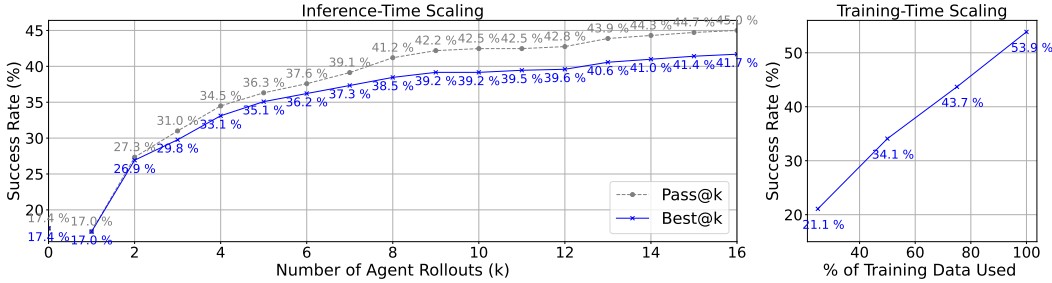

Figure 4: Scalable improvements of LLM agents in MedAgentGym. For inference-time scaling, we employ $T = 0$ for the initial rollout and $T = 0.6$ for the rest. For train-time scaling, we set $T = 0$.

RL optimizes the policy by favoring selected responses over rejected ones (Figure 3). We further consider a two-stage fine-tuning, initially warming up with SFT and subsequently refining with RL.

**Results: Offline RL (DPO).** Table 4 compares several post-training methods, revealing that simple SFT over successful trajectories significantly boosts performance on structured coding tasks, demonstrating its effectiveness in capturing structured coding patterns. Besides, DPO is particularly beneficial for optimizing open-ended task performance.

**Results: Online RL (PPO and GRPO).** We further consider online RL methods, including Proximal Policy Optimization (PPO) (Schulman et al., 2017) and Group Relative Policy Optimization (GRPO) (Shao et al., 2024b), to enable Med-Copilot to actively explore tasks and dynamically generate higher-quality training data through interaction. The evaluation module of Med-Copilot is employed to provide two reward signals: a correctness reward and a format reward, the latter indicating whether the generated output contains code blocks. As shown in Table 4, GRPO achieve markedly stronger performance, suggesting enhanced generalization capabilities in diverse biomedical scenarios compared with offline RL.

## 5.2 SCALING LLM AGENT IMPROVEMENTS WITH MEDAGENTGYM

**Verifier Training Setup.** In addition to directly training coding agents, MedAgentGym facilitates the development of an outcome-supervised reward model (ORM) to evaluate generated solutions effectively. Inspired by prior work (Cobbe et al., 2021; Pan et al., 2025), we formalize the verifier task as predicting the probability that a given trajectory successfully solves a coding task. Formally, we represent a trajectory as an interleaved sequence $\tau = [o_1, a_1, o_2, a_2, \cdots, o_n, a_n]$, $r \in [0, 1]$, where each observation $o_k$ comprises elements such as task descriptions, code execution results, and error feedback. We fine-tune a Qwen2.5-7B-Instruct model as a verifier with binary predictions 'YES' ($l_y$) or 'NO' ($l_n$), from which we compute success probability: $r = \exp(l_y)/(\exp(l_y) + \exp(l_n))$.

**Verifier Training Data.** We construct the verifier training dataset by combining two sets of trajectories originally sampled for agent training: (1) *off-policy trajectories*, consisting of 2,742 samples from gpt-4.1-mini; and (2) *on-policy trajectories*, comprising 2,939 samples generated by the agent. Combining both on- and off-policy trajectories, we ensure a balanced dataset of successful and unsuccessful trajectories, filtering to fit within a maximum context length of 32k tokens.

**Results: Inference and Training-Time Scaling.** We introduce two additional evaluation metrics: (1) *Pass@K*: the fraction of tasks solved by at least one trajectory from $K$ sampled attempts; and (2) *Best@K*: the fraction of accurately selects successful trajectories that actually solves the task from a set of candidate generations. Figure 4 (left) illustrates the performance scaling with increasing trajectory sampling. Pass@K significantly improves from 17.0% at $K = 1$ to 45.0% at 16, while

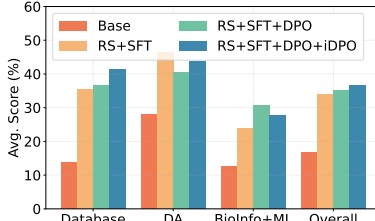

Figure 5: Self-Improvement

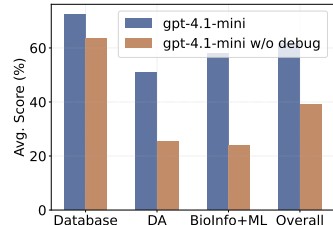

Figure 6: Effect of Debug

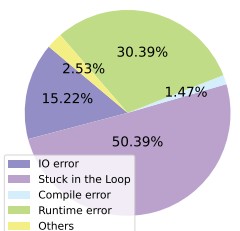

Figure 7: Error Types

Best@K shows steady advancement from 17.0% to 41.7%. The relatively small gap between metrics indicates that our trained verifier effectively identifies successful trajectories, unleashing its potential as a reward model for integration into advanced online RL frameworks. Figure 4 (right) examines agent performance as a function of increased training data volumes in SFT. We observe consistent performance improvements with greater training data availability, suggesting additional computational resources dedicated to sampling further trajectories are likely to yield continued performance gains.

## 5.3 MODEL PERFORMANCE SCALING WITH SELF-IMPROVEMENT

**Self-Improvement Training Setup.** Beyond expert-generated trajectories from `gpt-4.1-mini`, we also explore self-improvement by refining the model on its own outputs (`Qwen2.5-7B-Instruct`). We first apply rejection-sampling SFT: starting from `Qwen2.5-7B-Instruct`, we collect 1,000 successful trajectories and perform filtered behavior cloning on this set. We subsequently apply DPO (section 5.1) using on-policy preference pairs generated by the rejection sampling SFT checkpoint. Specifically, we sample eight rollouts per task, score them via the verifier in section 5.2, and form 4,298 pairs by contrasting the highest-scoring correct and lowest-scoring incorrect trajectories. Following Pang et al. (2024), we repeat this data collection and policy update cycle (iDPO) for further refinement, resampling trajectories and reconstructing preference pairs after each DPO update. In contrast to standard DPO, which performs a single offline preference-based update on a fixed dataset, iDPO alternates between on-policy data collection and DPO optimization so that the policy and training data co-evolve (*i.e.*, self-improvement).

**Results: Rejection Sampling (RS) and iDPO.** Figure 5 illustrates consistent performance gains across one SFT stage and two subsequent DPO stages. However, we observe diminishing returns over successive iterations. Initially, rejection sampling SFT significantly boosts performance by effectively capturing successful coding patterns. Subsequent DPO stages show smaller incremental improvements, reflecting the model's diminishing exploration space as it tackles increasingly challenging tasks, ultimately converging toward an approximate Nash equilibrium.

## 5.4 GENERALIZATION, ABLATION, AND ERROR ANALYSIS

**Results: External Evaluation.**
Table 5 summarizes external evaluation results on `MedAgentGym`. The external suites were intentionally chosen to stress different code-centric skills (*e.g.*, sequential SQL, hybrid text–table consistency, raw EHR time series, biostatistical power analysis), which induces markedly different agent trajectories (Figure 10(b)) and naturally yield lower absolute scores across all models, including proprietary baselines. Med-Copilot with SFT and DPO modestly improve performance on openended, reasoning-intensive tasks (*e.g.*, MIMIC-Extract). However, improvements remain limited, in-

Table 5: External test set results on `MedAgentGym`.

| Datasets (→) Base (↓) / Metrics (→) | EHR-SeqSQL SR | EHRCon SR | MIMIC-Extract Acc | N-PowerAI SR | Avg. Score |
|---|---|---|---|---|---|
| *API-based Proprietary LLMs[†] (for reference)* | | | | | |
| gpt-4o-mini (Hurst et al., 2024) | 50.80 | 23.20 | 2.67 | 16.03 | 26.03 |
| gpt-4o (Hurst et al., 2024) | 58.40 | 35.79 | 9.82 | 20.71 | 34.69 |
| gpt-4.1-mini (OpenAI, 2025a) | 70.60 | 52.40 | 5.62 | 25.66 | 43.20 |
| gpt-4.1 (OpenAI, 2025a) | 78.20 | **63.00** | 10.41 | 33.53 | 51.06 |
| gpt-o4-mini (OpenAI, 2025b) | **100.00** | 51.00 | **16.88** | **36.15** | **53.94** |
| *OSS LLMs* | | | | | |
| Qwen3-1.7B (Qwen, 2025a) | 33.60 | 17.20 | 1.90 | 14.72 | 16.86 |
| Qwen3-4B (Qwen, 2025a) | 44.80 | 26.20 | 4.59 | 19.30 | 23.72 |
| Qwen3-8B (Qwen, 2025a) | 52.00 | 31.40 | 6.82 | 20.12 | 27.59 |
| ▽HuatuoGPT-o1-7B (Chen et al., 2024) | 33.25 | 19.80 | 2.11 | 12.45 | 16.90 |
| Qwen2.5-7B-Inst (Yang et al., 2024a) | 42.20 | 27.20 | 1.34 | 11.66 | 20.60 |
| **Med-Copilot (SFT, 7B)** | 42.40 | 28.80 | 1.95 | 10.48 | 20.91 |
| **Med-Copilot (DPO, 7B)** | 43.40 | 23.00 | 2.14 | 14.82 | 20.84 |
| **Med-Copilot (PPO, 7B)** | 45.60 | 24.40 | 4.30 | 17.19 | 22.87 |
| **Med-Copilot (GRPO, 7B)** | 61.25 | 46.80 | 10.80 | 27.65 | 36.63 |
| Qwen3-14B (Qwen, 2025a) | 69.00 | 45.00 | 9.24 | 23.59 | 36.71 |
| Qwen2.5-Coder-14B-Inst (Hui et al., 2024) | 52.40 | 42.00 | 6.77 | 28.95 | 32.53 |
| Qwen2.5-14B-Inst (Yang et al., 2024a) | 46.40 | 39.20 | 4.51 | 21.57 | 27.92 |
| **Med-Copilot (DPO, 14B)** | 42.20 | 40.80 | 2.75 | 25.89 | 27.91 |
| **Med-Copilot (PPO, 14B)** | 66.40 | 43.70 | 7.15 | 32.01 | 37.32 |
| **Med-Copilot (GRPO, 14B)** | 72.80 | 56.60 | 14.91 | 43.77 | 47.02 |
| R1-Dis-Qwen-14B (Guo et al., 2025) | 56.00 | 40.80 | 2.37 | 17.60 | 29.19 |
| Qwen3-32B (Qwen, 2025a) | 64.80 | 54.40 | 12.17 | 31.26 | 42.16 |

dicating challenges in generalizing across specialized biomedical contexts. In particular, incorporating online RL optimization techniques, especially GRPO (Shao et al., 2024b), can effectively improve performance on unseen, out-of-distribution tasks. Specifically, Med-Copilot-14B (GRPO) achieves 47.02% on the external suite, a +19.10% gain over its backbone (27.92%). This significant gain on unseen distributions verifies that `MedAgentGym` instills transferable biomedical coding proficiency rather than memorizing training trajectories.

**Effect of Interactive Coding.** Figure 6 shows that removing debugging capabilities significantly decreases model performance across all tasks. Interactive coding mechanism in `MedAgentGym` substantially contributes to successful coding-based medical reasoning by enabling the model to effectively interpret and rectify execution errors.

**Error Analysis.** Figure 7 summarizes common error types encountered by the strongest evaluated LLM, `gpt-4.1`. Loop-related issues dominate, accounting for 50.39% of errors, where agents repeatedly execute the same action in the final turns, indicating difficulty in adapting or exploring alternative strategies. This highlights the need to promote effective exploration and enhance robustness in solving complex biomedical reasoning tasks. Additional experimental results, including cost analysis, case studies, and human studies, are available in appendix F.

## 6    CONCLUSION

We present `MedAgentGym`, an executable, privacy-preserving, and extensible training environment for scaling code-based biomedical reasoning in LLM agents. With 72K task instances across 129 categories, `MedAgentGym` enables comprehensive benchmarking of 29 proprietary and OSS LLMs for biomedical data science within a modular, decoupled architecture that supports flexibility and extensibility. `Med-Copilot` further demonstrates that systematic training and trajectory sampling with `MedAgentGym` improve coding proficiency for biomedical data science tasks. `MedAgentGym` has the potential to accelerate progress from structured medical information retrieval tasks toward more open-ended computational research questions in clinical research and biomedical discovery.

## ETHICS STATEMENT

We confirm that all authors read and will adhere to the ICLR Code of Ethics. This study uses only publicly available or credentialed deidentified datasets (*e.g.*, MIMIC-III and eICU) under their licenses or data use agreements. We do not redistribute data that require credentialed access; instead, we provide scripts to obtain and prepare such data. Licensing and access requirements for all datasets and associated code bases are summarized in Table 6, and privacy practices are detailed in appendix A.3. In particular, we followed the PhysioNet Credentialed Health Data Use Agreement for MIMIC-III and eICU and did not transfer any confidential patient data to third-party services. When using Microsoft Azure OpenAI services, we opted out of human review and followed the PhysioNet guidelines for responsible use.

## ACKNOWLEDGMENTS

RX and CY were partially supported by the internal funds and GPU servers provided by the Computer Science Department of Emory University, Emory Global Diabetes Center of the Woodruff Sciences Center, the US National Science Foundation under Award Numbers 2442172, 2312502, 2319449, and the US National Institutes of Health under Award Numbers K25DK135913, RF1NS139325, R01DK143456 and U18DP006922. GX was partially supported by the the US National Institutes of Health under Award Numbers R01GM140012, R01GM115473, U01CA249245, and CPRIT Awards under RP230330 and RP250561. YX was partially supported by the the US National Institutes of Health under Award Numbers U01AI169298, R35GM136375, and CPRIT RP240521. WS was partially supported by the internal funds and GPU servers provided by the Texas Advanced Computing Center (TACC), NVIDIA Academic Grant Program, Google Gemini Academic Program Award, and Thinking Machines Lab Tinker Research Grant.

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

## A  LIMITATIONS AND BROADER IMPACTS

### A.1  LIMITATIONS

**Resource Limitations.**  Although `MedAgentGym` demonstrates strong empirical performance improvement in a wide range of coding-aided biomedical reasoning tasks, several limitations remain. Firstly, `MedAgentGym` requires substantial computational resources for trajectory sampling, model fine-tuning, and iterative self-improvement procedures. Although we achieve significant improvements with relatively lightweight OSS LLMs, further scaling and advanced RL methods require increased computing infrastructures, limiting accessibility for resource-constrained research groups. Secondly, our current dataset size and trajectory collection are primarily constrained by computational budget rather than data availability, potentially limiting the full exploration of model scaling behavior. Thirdly, `MedAgentGym` primarily supports text and structured data modalities. Future extensions will incorporate multimodal biomedical data (*e.g.*, medical imaging, EEG, audio or video signals), enabling a richer and more comprehensive evaluation of multi-modal reasoning capabilities. Achieving effective multi-modal integration, however, presents significant challenges in data collection, curation, and standardized evaluation frameworks. Lastly, because of the substantial computational cost of API-based LLMs, we restrict ourselves to large-scale, execution-verified single-run evaluation under a fixed scaffold. We encourage future work to conduct more extensive uncertainty analyses and multi-run evaluations where resources allow.

**Data Decontamination.** We acknowledge the possibility of data contamination in the pre-training of proprietary LLMs. To mitigate this, we have taken several steps: (1) *Restricted access and credentialing*: We constructed many datasets from protected data that cannot be used for proprietary LLMs training (*e.g.*, MIMIC, EHRSHOT, EHR-SeqSQL, EHRCon, MIMIC-Extract) or very recent sources where possible (*e.g.*, MedAgentBench, BioDSBench). (2) *Newly curated samples*: For N-PowerAI, we manually curated samples rather than a public repository, effectively creating a private evaluation set. (3) *Rigorous N-Gram decontamination*: We performed n-gram overlap checks to eliminate direct contamination between our training and test splits.

### A.2  BROADER IMPACTS

**Potential Positive Societal Impacts.** `MedAgentGym` can significantly enhance the development of accessible, affordable, and privacy-preserving AI tools for clinical decision-making. Improved coding-based biomedical reasoning capabilities in open-source LLM agents (*e.g.*, `Med-Copilot`) have the potential to democratize access to advanced computational healthcare assistance, benefiting clinicians, researchers, and healthcare systems globally, particularly in resource-limited settings. The plug-and-play architecture also allows continuous adaptation to new medical knowledge and practices, fostering sustainable and community-driven innovation in healthcare technology.

**Potential Negative Societal Impacts.** Despite the benefits, the introduction and widespread deployment of sophisticated computational frameworks like `MedAgentGym` may unintentionally widen existing healthcare inequities. Institutions with limited computational resources (including both Microsoft Azure API service and high-performance computing clusters) or inadequate data infrastructure may struggle to access or fully benefit from these technological advancements, potentially exacerbating disparities in healthcare capabilities across regions or socioeconomic groups. Moreover, reliance on publicly available datasets may perpetuate existing biases due to uneven data representation, potentially disadvantaging underrepresented patient populations and rare disease conditions.

**Implications for Low-Resource Settings.** For institutions constrained to <10B OSS models, we recommend the following practices: **(1) Prioritize agentic fine-tuning over zero-shot prompting.** Zero-shot performance of small OSS models is modest (*e.g.*, Qwen3-8B at 30%), but agentic fine-tuning in `MedAgentGym` substantially closes this gap. In particular, GRPO lifts a 7B backbone from 16.89% to 62.17%, approaching the performance of commercial APIs (*e.g.*, gpt-4.1-mini at 61.72%). This suggests that small models are viable for deployment if they are post-trained, rather than used purely in zero-shot mode. **(2) Exploit released models and trajectories to reduce compute.** To lower the barrier to entry, we release `Med-Copilot-7B/14B` as ready-to-use, fine-tuned models that achieve state-of-the-art performance among OSS baselines, as well as >6K high-quality trajectories (successful and preference pairs). These resources allow practitioners to (i) adopt `Med-Copilot` directly, or (ii) perform lightweight SFT/DPO on top of their own backbones

without incurring the full cost of trajectory sampling and filtering. **(3) Leverage verifier-gated inference rather than larger models.** We recommend integrating the outcome verifier at deployment time. By sampling a small number of candidate solutions and selecting with the verifier (Best@K), practitioners can substantially improve reliability relative to single-shot decoding, effectively trading modest additional inference compute for large accuracy gains, without resorting to much larger or proprietary models. **(4) Specialize to the target biomedical data science task type.** The aggregate scores in Table 3 reflect performance over 129 heterogeneous categories across 8 datasets, whereas real deployments often focus on narrower workloads (*e.g.*, clinical SQL, risk calculations, or bioinformatics pipelines). Fine-tuning a small model on the subset of MedAgentGym that matches the intended application typically yields higher task-specific performance than the overall average suggests, and is more compute-efficient than aiming for a fully generalist agent.

## A.3 PRIVACY STATEMENTS

Table 6: Data Access and License Information of 12 datasets in MedAgentGym. "Custom" represents additional dataset- or task-specific license and data access requirements (*e.g.*, DUA or credentials).

| Dataset | Data License | Data Access | Code License | Code Access |
|---|---|---|---|---|
| *Training and Internal Validation (In-Distribution)* | | | | |
| MIMIC-III (Johnson et al., 2016; Lee et al., 2022) | Custom | MIMIC-III on PhysioNet | CC-BY-4.0 | MIMIC-III on EHRSQL |
| eICU (Pollard et al., 2018; Lee et al., 2022) | Custom | eICU on PhysioNet | CC-BY-4.0 | eICU on EHRSQL |
| TREQS (Wang et al., 2020a) | Custom | MIMIC-III on PhysioNet | MIT | TREQS on GitHub |
| MedCalcBench (Khandekar et al., 2024) | CC-BY-SA 4.0 | MedCalcBench | Public | MedCalcBench on GitHub |
| MedAgentBench (Jiang et al., 2025b) | MIT | MedAgentBench (FHIR Server) | MIT | MedAgentBench on GitHub |
| BioCoder (Tang et al., 2024a) | CC-BY-4.0 | BioCoder on Huggingface | N/A | BioCoder on GitHub |
| BioDSBench (Wang et al., 2024b) | MIT | BioDSBench | MIT | BioDSBench on GitHub |
| EHRSHOT (Wornow et al., 2023a) | Custom | EHRShot (Standford) | Apache | EHRSHOT on Github |
| *External Validation (Out-of-Distribution)* | | | | |
| EHR-SeqSQL (Ryu et al., 2024) | Custom | MIMIC-III on PhysioNet | N/A | EHR-SeqSQL on GitHub |
| EHR-Con (Kwon et al., 2024) | Custom | MIMIC-III on PhysioNet | MIT | EHR-Con on GitHub |
| MIMIC-Extract (Wang et al., 2020b) | Custom | MIMIC-III on PhysioNet | MIT | MIMIC-Extract on GitHub |
| N-PowerAI (Ruan et al., 2025) | N/A | N-Power AI Supp. Mat. | N/A | N-Power AI on Webpage |

**Data Privacy and Licensing.** We carefully curated MedAgentGym with strict adherence to ethical standards, using publicly available datasets or datasets with appropriate privacy protections and anonymizations. Table 6 lists the access requirements for the 12 datasets in MedAgentGym and the code base for data processing or task implementation. We explicitly designed isolated Docker environments to ensure data privacy and security. Nevertheless, ethical usage of our methods and models in clinical settings requires rigorous validation, transparency in limitations, and close collaboration with healthcare professionals. We encourage responsible deployment, emphasizing human oversight, continuous evaluation, and clear communication of model capabilities and uncertainties to mitigate ethical and practical risks.

**LLM Usage Statement.** In compliance with the PhysioNet Credentialed Health Data Use Agreement (version 1.5.0)[4], we strictly prohibit transferring confidential patient data (*e.g.*, MIMIC-III and eICU) to third-party entities, including external online services and APIs. To responsibly utilize the Azure OpenAI Service, we adhere closely to PhysioNet's guidelines on responsible GPT usage[5]. Specifically, we have opted out of the human review process by completing the Azure OpenAI Additional Use Case Form[6], thereby ensuring no third-party entity accesses or processes sensitive patient information. We consistently monitor our data handling practices and strictly adhere to applicable guidelines and privacy regulations, maintaining the highest ethical standards in our research and operations.

## B ADDITIONAL RELATED WORKS

**Medical Reasoning Models.** Recent advancements have substantially improved biomedical reasoning capabilities of LLMs through RL (Huang et al., 2025b; Lai et al., 2025; Zhang et al., 2025b; Jiang et al., 2025a; Wu et al., 2025a; Chen et al., 2024; Lan et al., 2025; Wang et al., 2025a; Li et al., 2025; Zhang et al., 2025b; Miao et al., 2025; Jin et al., 2025; Yu et al., 2025; Zhi et al., 2025; Liu

---

[4]https://physionet.org/about/licenses/physionet-credentialed-health-data-license-150/

[5]https://physionet.org/news/post/gpt-responsible-use

[6]https://aka.ms/oai/additionalusecase

et al., 2025a). For example, M1 (Huang et al., 2025b) improves by distilling knowledge from the reasoning traces generated by DeepSeek-R1 (Guo et al., 2025). MedS3 (Jiang et al., 2025a) employs Monte Carlo Tree Search (MCTS) to generate rule-verifiable reasoning trajectories and employs process-reward models to select optimal reasoning paths during inference. Similarly, HuatuoGPT-o1 (Chen et al., 2024) and ClinicalGPT-R1 (Lan et al., 2025) integrate domain-specific verifiers to guide RL fine-tuning processes for improved clinical reasoning. Extending beyond language modeling, Med-R1 (Lai et al., 2025) and MedXpertQA (Zuo et al., 2025) adapt RL methodologies to vision-language models, effectively addressing medical visual question answering tasks. Despite these developments, current medical reasoning models predominantly target natural language-based reasoning, with limited attention given to coding-intensive scenarios common in biomedical research and clinical practice.

**Medical Reasoning Benchmarks.** Most existing medical reasoning benchmarks focus primarily on evaluating LLM performance through closed-form medical QA tasks (Pal et al., 2022; Jin et al., 2021; 2019; Tsatsaronis et al., 2015; Tang et al., 2025; Xiong et al., 2024; Arora et al., 2025). In addition, AgentClinic (Schmidgall et al., 2024) further evaluates diagnosis prediction within simulated clinical scenarios, while MedHELM (HAI@Stanford, 2025) provides comprehensive evaluations in various medical NLP tasks. Despite these extensive benchmarking efforts, existing benchmarks – including recent concurrent works such as MedAgentBoard (Zhu et al., 2025), HealthBench (Arora et al., 2025), and MedCaseReasoning (Wu et al., 2025b) – typically focus on evaluation scenarios, with limited emphasis on dedicated training environments aimed at systematically improving medical reasoning capabilities (Thapa et al., 2025), especially within coding-intensive and interactive medical scenarios.

**Medical Agents (Coding).** Recent advances have demonstrated that LLMs exhibit strong capabilities in medical reasoning and planning leveraging extensive biomedical knowledge (Singhal et al., 2023; Moor et al., 2023; Liévin et al., 2024), fueling increased interest in developing LLM-based autonomous agents tailored specifically for medical tasks (Jin et al., 2024; Gao et al., 2025; Li et al., 2024a; Liao et al., 2024; Tang et al., 2024b; Kim et al., 2024). In particular, LLM-based agents have shown promise in specialized computational tasks, including querying EHR databases (Shi et al., 2024b), performing bio-statistical calculations (Ruan et al., 2025), and conducting bioinformatics analyses (Tang et al., 2024a; Wang et al., 2024b; Tayebi Arasteh et al., 2024). As shown in Figure 8, integrating coding capabilities into LLM-based agents further enhances performance on tasks traditionally approached

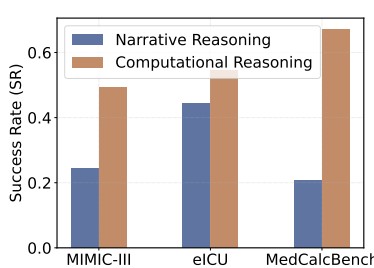

Figure 8: Coding empowers computational medical reasoning (w/ `gpt-4-turbo`).

through natural language reasoning (*e.g.*, MIMIC-III, eICU (Lee et al., 2022)), as well as numerical and rule-based medical reasoning (*e.g.*, MedCalcBench (Khandekar et al., 2024)). However, existing coding-based medical agents rely primarily on prompt engineering without systematic improvement, limiting their robustness and scalability when addressing complex and diverse coding tasks in real-world biomedical scenarios. In contrast, `MedAgentGym` specifically targets reasoning-intensive coding tasks by introducing a unified, scalable, and interactive training environment that systematically improves the coding-based medical reasoning capabilities of LLM agents.

**Medical Agent Training Environments.** To advance medical agents with narrative reasoning, AgentClinic (Schmidgall et al., 2024) and AgentHospital (Li et al., 2024b) simulate hospital workflows focused on diagnostic tasks, while MediQ (Li et al., 2024c) offers interactive simulations designed for medical information retrieval. Beyond medicine, specialized environments have emerged for systematically evaluating and improving LLM agents across diverse tasks (Zhao et al., 2025; Wang et al., 2025e), such as software engineering (Pan et al., 2025; Yang et al., 2024b; 2025), reasoning (Stojanovski et al., 2025), web browsing (Drouin et al., 2024), agent planning and collaboration (Xi et al., 2024; Shao et al., 2024a), data science (Guo et al., 2024; Jing et al., 2025; Zhang et al., 2025a; 2024), machine learning engineering (Nathani et al., 2025; Huang et al., 2023; Chan et al., 2024; Tang et al., 2023), automated research (Kang & Xiong, 2024; Schmidgall & Moor, 2025; Schmidgall et al., 2025), and scientific discovery (Team et al., 2025; Yuan et al., 2025). Inspired by these interactive training frameworks, `MedAgentGym` uniquely targets real-world biomedical scenarios, aiming to rigorously benchmark and systematically enhance coding-based biomedical reasoning capabilities of LLM agents. Unlike general coding agent benchmarks that primarily target software

engineering tasks (Jimenez et al., 2024; Yang et al., 2025), MedAgentGym emphasizes biomedical coding reasoning, requiring integration of clinical knowledge and domain-specific data formats (*e.g.*, EHRs, lab reports, biological sequences) within executable environments.

## C  TASK AND DATA DETAILS

### C.1  OVERVIEW

We refer a task as coding-based biomedical reasoning when LLM agents write and run code whose execution yields a verifiable outcome in biomedical data science. This definition allows us to objectively verify the results while preserving the steps that agents actually take, allowing for training and analysis at the trajectory level.

**Biomedical Application Category.** MedAgentGym spans multiple biomedical subdomains, including *Database queries* (DB, including MIMIC-III, eICU, TREQS, EHR-SeqSQL, and EHRCon), *Data Analytics* (DA, including MedCalcBench and MedAgentBench), *Bioinformatics* (Bioinfo, including BioCoder, BioDSBench, N-PowerAI), and *Machine Learning* (ML, including EHRSHOT and MIMIC-Extract).

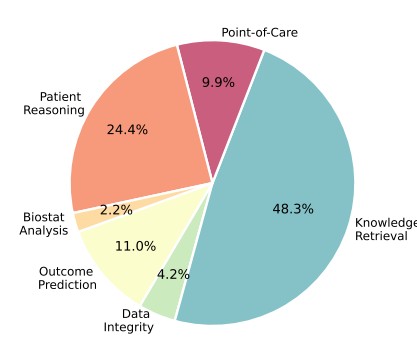

Figure 9: Diversity analysis.

Figure 9 illustrates the diverse task distribution within MedAgentGym. Consider a clinician identifying patients at risk for sepsis from EHR data, a task requiring not only understanding of sepsis criteria but also SQL queries to extract relevant laboratory values, temporal logic to track patient trajectories, and statistical methods to validate findings. Similarly, researchers analyzing multi-omics data must integrate biological knowledge with bioinformatics algorithms and computational pipelines. These scenarios exemplify the core challenge of biomedical data science: operationalizing medical expertise through executable code, where domain knowledge alone proves insufficient without corresponding computational implementation.

**In- & Out-of-Distribution.** We further categorize tasks in MedAgentGym into *in-* and *out-of-distribution*, facilitating a rigorous evaluation of model generalization and adaptability. To highlight intrinsic differences between these distributions, Figure 10(b) shows the distribution of sampled code trajectories. The resulting visualization demonstrates significant divergence in trajectory complexity, interaction frequency, and required code refinement steps between in-distribution and out-of-distribution tasks, underscoring the challenges posed by novel biomedical reasoning contexts.

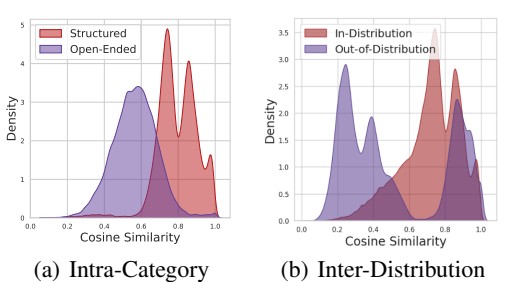

(a) Intra-Category          (b) Inter-Distribution

Figure 10: Similarity analysis

**Computational Task Category.** *Structured tasks* primarily include database query scenarios, such as those from MIMIC-III, eICU, TREQS, EHR-SeqSQL, EHRCon, and MedCalcBench (rule- or equation-based), which require precise formulation of executable queries against structured EHR data. *Open-ended tasks* include biomedical data analysis and medical coding scenarios drawn from datasets such as MedAgentBench, BioCoder, BioDSBench, EHRSHOT, MIMIC-Extract, and N-PowerAI, demanding nuanced and flexible code generation for complex analysis, statistical reasoning, or clinical decision-making.

Specifically, we evaluate LLMs across eight biomedical coding domains: (1) clinical database querying (MIMIC-III, eICU, TREQS, EHRseqSQL), (2) clinical note analysis (EHRcon), (3) medical computation (MedCalcBench), (4) health information technology (MedAgentBench), (5) biomedi-

cal software engineering (Biocoder), (6) biomedical data analysis (BioDSBench), (7) biostatistics (NPowerAI), and (8) ML-based predictive modeling (EHRSHOT, MIMIC-Extract).

## C.2 TRAINING AND INTERNAL TESTING (IN-DISTRIBUTION) DATASET DETAILS

**EHRSQL: MIMIC-III and eICU.** EHRSQL (Lee et al., 2022) comprises text-to-SQL tasks that leverage electronic health records from MIMIC-III (Johnson et al., 2016) and eICU (Pollard et al., 2018). They evaluate the ability of LLMs (and agents) to translate clinical questions posed by healthcare professionals into executable SQL queries. This includes handling complex queries involving temporal logic and conditional abstention.

**TREQS.** TREQS (Wang et al., 2020a) is a text-to-SQL benchmark tailored specifically to clinical question answering using the MIMIC-III dataset. It emphasizes generating accurate SQL queries from template-based natural language questions against a simplified schema comprising five core tables, with an emphasis on large result-set handling.

**MedCalcBench.** MedCalcBench (Khandekar et al., 2024) provides a structured evaluation of clinical calculation capabilities in LLMs. Each instance poses a patient-specific clinical scenario requiring precise medical calculations such as clinical scores or medication dosages, accompanied by expert-curated stepwise solutions for validation.

**MedAgentBench.** MedAgentBench (Jiang et al., 2025b) is a simulated EHR environment designed to evaluate LLM-driven clinical workflows. It features realistic patient scenarios across ten task categories, requiring agents to perform clinical reasoning, EHR querying via FHIR interfaces, and clinical decision support.

**BioCoder.** BioCoder (Tang et al., 2024a) assesses the capability of LLMs to generate accurate bioinformatics code solutions. It comprises practical coding challenges derived from authentic bioinformatics software, requiring the generation and verification of functionally correct Python methods.

**BioDSBench.** BioDSBench (Wang et al., 2024b) evaluates LLM proficiency in biomedical data science coding tasks, involving the generation of Python or R code to replicate analytical workflows derived from actual biomedical research studies. Tasks span statistical analyses, data manipulations, and visualization routines.

**EHRSHOT.** EHRSHOT (Wornow et al., 2023a) benchmarks LLMs on few-shot clinical prediction tasks leveraging real-world, longitudinal, deidentified EHR data. It focuses on rapid adaptation to tasks such as risk prediction and forecasting clinical outcomes given limited labeled examples.

## C.3 EXTERNAL EVALUATION (OUT-OF-DISTRIBUTION) DATASET DETAILS

**EHR-SeqSQL.** EHR-SeqSQL (Ryu et al., 2024) extends text-to-SQL evaluation to sequential, multi-turn interactions, emulating realistic clinical dialogues. Tasks require maintaining context across multiple SQL queries, assessing LLM capability in handling compositional and contextual reasoning.

**EHRCon.** EHRCon (Kwon et al., 2024) involves assessing clinical note consistency with structured EHR records, focusing on identifying discrepancies. It serves as a verification task requiring precise alignment between unstructured clinical text and corresponding database entries.

**MIMIC-Extract.** MIMIC-Extract(Wang et al., 2020b) provides structured, preprocessed time-series patient data derived from the MIMIC-III dataset, used in clinical predictive modeling such as mortality risk or intervention prediction, enabling standardized assessments of time-series reasoning capabilities.

**N-PowerAI.** N-PowerAI (Ruan et al., 2025) evaluates LLM capabilities in performing statistical sample-size and power analyses for clinical trial design. It requires multi-step statistical reasoning and the generation of precise numeric results corresponding to various clinical scenarios.

## C.4 TRAIN-TEST SET SPLIT

For datasets that provide predefined training, validation, and test splits, we combine the training and validation subsets into a single unified training set and retain the original test subset exclusively for

evaluation. In cases where datasets lack predefined splits, we randomly allocate 50% of the instances to training, assigning the remaining 50% to the test set. For tasks containing more than 1000 samples in both training and test sets, we create a lighter subset through downsampling to support efficient leaderboard-based training and evaluation. Specifically, we leverage task-specific metadata to perform uniform sampling within each fine-grained category, thereby maintaining diversity, ensuring balanced representation, and preserving the original data distribution.

## C.5  DATA PRE-PROCESSING DETAILS

Rather than a simple concatenation of existing benchmarks, MedAgentGym transforms and unifies heterogeneous biomedical datasets into a single executable, Docker-isolated environment with standardized JSON I/O, natural-language error grounding, and execution-verified tasks, together with multi-threaded trajectory sampling and an outcome-supervised verifier to support agentic RL training. Dataset-specific transformations are detailed as follows:

### C.5.1  STRUCTURED TASKS

For database querying related datasets, including **MIMIC-III**, **eICU**, **TREQS**, and **EHR-SeqSQL**, each task instance is structured into a JSON format comprising: (1) the contextual description and the corresponding natural-language query, (2) the ground-truth SQL query, and (3) the resulting answer from the database execution. Instances yielding null results upon SQL execution, indicating the absence of a valid answer, are excluded from the dataset.

For **EHRCon**, we organize the data into structured databases that link patient records through hospital admission IDs, complemented by a separate database containing associated clinical notes. Each task is formulated as a JSON object consisting of: (1) admission ID, (2) relevant medical terminology, (3) count of detected inconsistencies, and (4) a binary indicator denoting the presence or absence of inconsistencies.

For **MedCalcBench**, each instance initially consists of a patient note, a specific medical calculation query, a ground-truth answer, and a detailed step-by-step solution. To accurately evaluate the coding capabilities of LLM agents without direct guidance, we remove all intermediate calculation hints, presenting only the patient note and the calculation query for model inference.

For **N-PowerAI**, statistical analysis tasks are augmented through attribute substitution. Specifically, each original instance is expanded 100-fold by systematically replacing an attribute with a randomly chosen equivalent from a predefined valid range, preserving the integrity and interpretability of the statistical context. Each augmented instance includes recalculated values for sample size (N) and statistical power, stored systematically within JSON-formatted records.

### C.5.2  OPEN-ENDED TASKS

**MedAgentBench** instances require LLM agents to follow natural-language instructions to perform tasks within a FHIR-compliant interactive medical environment. We retain original instructions, solutions, and Medical Record Numbers (MRNs). To derive verifiable evaluation signals, we execute the provided ground-truth on the server-side environment to obtain authoritative reference answers.

**BioCoder** tasks require implementing biostatistics algorithms or addressing scientific programming challenges. Each instance comprises a problem description, context-specific code, test cases, and expected outputs. While evaluation datasets already contain all necessary components, training instances initially lack context-specific code and test cases. To address this gap, we employ the o3-mini model to auto-generate relevant context code and corresponding test cases based on provided ground-truth functions. Generated functions undergo rigorous validation via a code interpreter, retaining only verified, error-free instances. Additionally, we exclusively utilize the Python-based subset of BioCoder, deferring the JavaScript subset for subsequent integration.

**BioDSBench** instances involve biomedical data analysis tasks derived from real-world datasets. Features are systematically organized into directories by task, with each task's description and reference Python implementation captured within JSON structures.

For datasets dedicated to predictive model development (*e.g.*, **EHRSHOT** and **MIMIC-Extract**), initial features are provided in pre-processed form but necessitate additional table joining, filtering,

and integration to produce final training inputs. While labels accompany these tasks, explicit reference Python implementations are not provided, as evaluation metrics directly measure the accuracy of model predictions on predefined test subsets. Distinct subsets of training, validation, and testing data and labels are explicitly maintained and separately utilized for both training and evaluation phases.

### C.6 SAMPLED TRAJECTORY DETAILS

Table 7 details the proportion of action types (section 4.1) in trajectories. Structured tasks predominantly involve data retrieval (over 50%) from databases or resources, complemented by coding and debugging steps. In contrast, open-ended tasks require significant coding and debugging efforts due to diverse question types, often necessitating terminal interactions to install specialized biomedical packages. Although `MedAgentGym` contains extensive training data and allows repeated sampling, the current trajectory count primarily reflects computational budget constraints. Specifically, Figure 4 (right) demonstrates consistent performance improvements with increasing training data volume, indicating that expanded trajectory sampling through additional computational resources would yield further gains.

Table 7: Trajectory Composition (%).

| Actions ($\rightarrow$) | request info | terminal | code | debug |
|---|---|---|---|---|
| MIMIC-III | 71.07 | 0 | 28.84 | 0.08 |
| eICU | 72.17 | 0 | 27.13 | 0.70 |
| TREQS | 64.27 | 0 | 35.54 | 0.19 |
| MedCalc. | 0 | 0 | 74.91 | 25.09 |
| **Structured** | **51.88** | **0** | **41.61** | **6.52** |
| MedAgent. | 0 | 0 | 100 | 0 |
| BioCoder | 0 | 0.29 | 96.11 | 3.60 |
| BioDS. | 0 | 6.30 | 87.60 | 6.90 |
| EHRSHOT | 0 | 0.43 | 59.43 | 40.14 |
| **Open-ended** | **0** | **1.76** | **85.79** | **12.46** |
| `MedAgentGym` | **32.71** | **0.14** | **57.11** | **10.04** |

## D BASELINE DETAILS

We include additional details of the coding and medical domain-specific LLMs:

- **Qwen2.5-Coder-Instruct** (Hui et al., 2024) is derived from the Qwen2.5 series and further fine-tuned explicitly on large-scale coding datasets and coding-specific instruction sets. This targeted training substantially enhances their capabilities in code generation, debugging, and programmatic reasoning, outperforming general-purpose models of similar scale on coding tasks.

- **Seed-Coder-8B-Reasoning** (Seed et al., 2025) is an 8B-parameter open-source coding LLM optimized for code generation, leveraging Long-Chain-of-Thought (LongCoT) reinforcement learning to improve multi-step code reasoning.

- **medgemma-4b-it** (gemma-3-4b-pt) (Google, 2025) is a medical-domain variant based on gemma architecture and fine-tuned specifically on medical QA and instruction datasets, which provide strong capabilities for medical reasoning and question answering.

- **HuatuoGPT-o1-7B** (Qwen2.5-7B-Instruct) (Chen et al., 2024), built on the Qwen2.5-7B architecture, is extensively fine-tuned in clinical reasoning datasets via PPO with verifier-based rewards to enhance complex reasoning capabilities. Specifically, it incorporates a medical-specific verifier model that guides the generation of complex reasoning trajectories. HuatuoGPT-o1-7B excels in medical reasoning tasks by explicitly generating intermediate reasoning steps that facilitate iterative refinement and introspective evaluation. We also evaluate **HuatuoGPT-o1-72B** (Qwen2.5-72B) to provide a more equitable and rigorous comparison with large-scale LLMs.

- **m1-7B-23K** (Qwen2.5-7B-Instruct) (Huang et al., 2025b) is fine-tuned on approximately 23,000 rigorously curated medical QA examples, significantly enhancing its domain-specific knowledge and reasoning capabilities.

- **MedReason-8B** (Llama-3.1-8B-Instruct) (Wu et al., 2025a) is fine-tuned for medical questions-answering and clinical reasoning tasks. Its training emphasizes the generation of step-by-step rationales, enabling robust performance on medical reasoning and diagnostic tasks.

- **Baichuan-M1-14B-Instruct** (Wang et al., 2025a) is a 14B medical LLM pre-trained from scratch on approximately 20 trillion tokens of medical domain-specific content and high-quality general text. It integrates specialized modeling across over 20 medical specialties with advanced architectural modifications enhancing context understanding and long-sequence reasoning.

- **Baichuan-M2-32B** (Dou et al., 2025) is a 32B medical LLM pre-trained from scratch on large-scale medical corpora and high-quality general text, with architectural and training adaptations for multi-specialty clinical reasoning and long-context understanding. We use it as a representative large medical-domain baseline.

# E    IMPLEMENTATION DETAILS

**Evaluation Metrics.** Following existing agent benchmarks (Liu et al., 2023), we adopt *success rate (SR)* as the primary evaluation metric. For *database, data science, and bioinformatics* tasks with explicit ground truths, we compare LLM-generated code execution outputs with reference solutions using exact match. For open-ended *ML* tasks in clinical decision support, we measure performance using *accuracy (Acc)* across provided test cases. Note that these code generation tasks inherently have infinite solution spaces, unlike traditional classification problems with bounded solution spaces (*e.g.*, even random guessing can yield around 50% accuracy in binary classification). The *overall score* is computed by averaging performance across tasks in test sets of `MedAgentGym` (leaderboard), providing a comprehensive evaluation of coding-based biomedical reasoning capabilities within `MedAgentGym`.

**Experimental Setup Details.** We limit interactions to a maximum of 15 turns per session, providing agents full access to interaction histories and constraining runtime to 120 seconds per session. Input tokens are capped at $32,768$, with output limited to $8,192$ tokens per round. We use Python 3.10 as the primary language for agent-code execution due to its modular design and suitability for biomedical computations. To enable interactive feedback (section 3.3), we employ a rule-based parser converting LLM outputs to JSON, facilitating seamless code execution, and utilize `gpt-4.1-mini` to translate execution errors into grounded explanations. We configure all baseline LLMs following established best practices for reproducibility. Specifically, instruction-following LLMs are configured with a temperature of zero, while reasoning models use a temperature of 0.6. For all experiments with `Qwen-3` series, we switch to thinking mode for optimal performance under complex reasoning scenarios (*e.g.*, logic, math, and coding).

**SFT.** For SFT experiments, smaller models (up to 8B parameters) are trained using eight NVIDIA A100 GPUs, whereas the 14B-parameter model is trained on eight NVIDIA H200 GPUs. We utilize the AdamW optimizer (Loshchilov & Hutter, 2017) with a learning rate of $1e-4$. The training batch size is set to 8, and the maximum input token length per batch is configured to 40,000 tokens.

**DPO.** DPO experiments are conducted using the same hardware configurations as SFT experiments. We employ the AdamW optimizer with a reduced learning rate of $5e-6$. Training utilizes a batch size of 64 and a KL-divergence coefficient ($\beta$) of 0.1 to regulate the divergence from the initial policy.

**PPO & GRPO.** PPO and GRPO experiments are conducted using the same hardware configurations as SFT experiments. All online RL experiments are conducted using VeRL framework (Sheng et al., 2025). We integrate the VeRL package and dependencies inside the `Med-Copilot` docker image to enable communication between the reward functions and the evaluation module. PPO and GRPO training is performed with a batch size of 128 and a learning rate of $1 \times 10^{-5}$. The temperature parameter during model rollout is consistently set to 0.6. Throughout training, the coefficient for the KL divergence regularization term is fixed at $\beta = 1 \times 10^{-3}$.

# F    ADDITIONAL EXPERIMENTAL RESULTS

## F.1    CODE QUALITY AND EFFICIENCY

For a comprehensive evaluation, we further report additional evaluation metrics on code quality and efficiency, including (1) **number of turns** for interaction effectiveness, (2) cyclomatic **complexity** for code complexity, (3) **maintainability** index for code readability, and (4) **line-of-code (loc)** and (5) **logical line-of-code (lloc)** for code efficiency (Table 8). Comparing different tasks (take gpt-4.1 for example), we observe that machine learning tasks such as EHRSHOT involve significantly higher complexity and longer code. Comparing different models (averaged across datasets), we observe that advanced closed-source models generate more complex and longer code; after training, `Med-Copilot` produces structurally efficient and more maintainable code compared to backbone models.

Table 8: Additional evaluation on code quality and efficiency for in- and out-of-distribution tasks.

| Datasets (→) | MIMIC. | eICU | TREQS | MedCalc. | MedAgent. | BioCoder | BioDS. | EHRSHOT | ID Avg. | EHR-SeqSQL | EHR-Con. | MIMIC-Extract | Npower-AI | OOD Avg. |
|---|---|---|---|---|---|---|---|---|---|---|---|---|---|---|
| *gpt-4.1 (2025-04-14)* | | | | | | | | | | | | | | |
| #turns | 25.91 | 26.59 | 20.65 | 10.73 | 17.28 | 22.08 | 21.75 | 8.71 | 19.21 | 25.83 | 38.97 | 10.42 | 22.64 | 20.83 |
| complexity | 0.01 | 0.06 | 0.01 | 4.09 | 0.23 | 7.77 | 0.17 | 20.97 | 4.16 | 0.03 | 0.11 | 20.76 | 0.04 | 4.49 |
| maintainability | 95.14 | 95.99 | 96.62 | 88.38 | 91.04 | 68.20 | 92.67 | 56.24 | 85.54 | 94.17 | 92.65 | 64.82 | 96.94 | 86.03 |
| loc | 9.26 | 9.67 | 4.17 | 19.00 | 18.89 | 24.82 | 28.97 | 144.69 | 32.43 | 9.44 | 12.45 | 129.76 | 5.24 | 34.52 |
| lloc | 5.86 | 6.33 | 3.00 | 15.20 | 10.79 | 21.84 | 16.44 | 110.51 | 23.75 | 6.21 | 8.90 | 117.54 | 3.73 | 26.93 |
| *gpt-4.1-mini (2025-04-14)* | | | | | | | | | | | | | | |
| #turns | 19.66 | 19.90 | 16.35 | 9.18 | 19.20 | 23.08 | 16.53 | 22.60 | 18.31 | 23.42 | 32.00 | 12.40 | 18.78 | 19.34 |
| complexity | 0.02 | 0.04 | 0.01 | 3.51 | 0.03 | 7.30 | 0.26 | 19.85 | 3.88 | 0.01 | 0.01 | 19.88 | 0.02 | 4.22 |
| maintainability | 95.62 | 96.06 | 98.93 | 87.01 | 94.43 | 69.43 | 92.54 | 57.77 | 86.47 | 92.18 | 96.91 | 54.63 | 97.72 | 86.13 |
| loc | 16.49 | 14.47 | 6.85 | 23.37 | 13.08 | 25.98 | 28.17 | 171.69 | 37.51 | 29.50 | 12.88 | 134.55 | 6.78 | 40.10 |
| lloc | 8.05 | 7.22 | 3.68 | 17.58 | 7.92 | 20.78 | 15.40 | 119.58 | 25.03 | 13.67 | 5.88 | 118.90 | 4.91 | 28.35 |
| *Qwen2.5-7B-Instruct* | | | | | | | | | | | | | | |
| #turns | 17.23 | 14.81 | 12.38 | 5.98 | 14.39 | 25.42 | 9.31 | 15.33 | 14.36 | 20.60 | 26.91 | 10.92 | 18.44 | 15.85 |
| complexity | 0.02 | 0.02 | 0.01 | 4.41 | 0.01 | 4.78 | 0.30 | 11.09 | 2.58 | 0.01 | 0.02 | 11.85 | 0.01 | 2.70 |
| maintainability | 96.54 | 96.02 | 98.58 | 82.65 | 80.20 | 81.67 | 95.66 | 85.75 | 85.75 | 93.38 | 94.12 | 64.98 | 96.68 | 86.22 |
| loc | 16.81 | 17.07 | 8.72 | 28.54 | 49.09 | 20.81 | 22.00 | 137.85 | 37.61 | 24.54 | 11.98 | 121.98 | 8.52 | 38.89 |
| lloc | 7.52 | 8.23 | 4.38 | 18.09 | 25.34 | 15.46 | 11.79 | 90.58 | 22.67 | 16.78 | 6.25 | 105.56 | 6.14 | 26.06 |
| *Med-Copilot (7B)* | | | | | | | | | | | | | | |
| #turns | 20.74 | 17.80 | 14.31 | 7.86 | 16.24 | 28.97 | 16.80 | 29.73 | 19.06 | 28.76 | 30.87 | 18.76 | 26.32 | 21.25 |
| complexity | 0.01 | 0.01 | 0.01 | 3.81 | 0.01 | 5.08 | 0.04 | 18.66 | 3.45 | 0.01 | 0.04 | 18.42 | 0.02 | 3.81 |
| maintainability | 94.58 | 95.01 | 98.49 | 83.76 | 82.64 | 81.40 | 97.68 | 62.47 | 87.00 | 94.54 | 98.81 | 72.82 | 98.76 | 88.30 |
| loc | 21.58 | 19.88 | 12.00 | 25.42 | 53.67 | 24.76 | 17.16 | 141.50 | 39.50 | 32.14 | 14.58 | 120.97 | 8.71 | 40.91 |
| lloc | 9.95 | 9.10 | 5.73 | 17.74 | 26.26 | 17.82 | 9.11 | 95.97 | 23.96 | 16.88 | 7.43 | 110.45 | 5.50 | 27.38 |

## F.2 ABLATION STUDY: EFFECT OF PRE-DEFINED TOOLSET

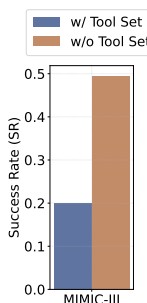

Figure 11: Effect of toolset.

Figure 11 compares the performance of GPT-4-based agents on the MIMIC-III dataset with and without predefined toolsets integrated into our agent scaffold. This illustrates our agent scaffold's ability to flexibly accommodate external tools. Interestingly, despite providing a set of predefined tools, including functions for database loading, data filtering, value retrieval, arithmetic calculations, date computations, and SQL execution (see additional details of toolset in Shi et al. (2024b)), we observe a surprising decline in agent performance. It suggests that the LLM agent inherently generates more flexible and contextually appropriate code when unencumbered by predefined function constraints, aligning with the observations reported by (Qian et al., 2025; Qiu et al., 2025).

## F.3 COST ANALYSIS

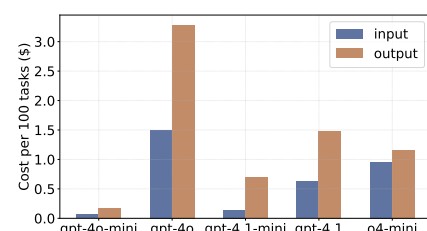

Figure 12: Cost information.

Table 9 summarizes input and output token statistics for various API-based proprietary LLMs evaluated on datasets within MedAgentGym. Notably, the input and output token lengths per query vary significantly across models and tasks. Among these models, gpt-4.1-mini achieves relatively low average input and moderate output token counts, which implies more efficient token utilization during inference compared to larger variants such as gpt-4o and gpt-o4-mini. Conversely, gpt-o4-mini incurs higher average input costs. Figure 12 presents the API cost per 100 tasks. Overall, smaller GPT variants (*e.g.*, gpt-4.1-mini and gpt-4o-mini) offer superior token-efficiency, translating into lower computational and API costs without substantial compromise in performance, demonstrating their effectiveness as cost-efficient solutions for large-scale biomedical reasoning applications.

## F.4 STRUCTURED AND OPEN-ENDED TASKS

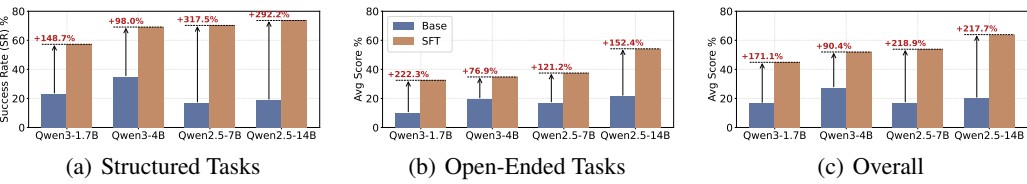

(a) Structured Tasks     (b) Open-Ended Tasks     (c) Overall

Figure 13: Med-Copilot SFT performance on MedAgentGym across various backbone LLMs.

Table 9: Statistics of input and output tokens per question for API-based commercial LLMs.

| Datasets (→) | MIMIC. | eICU | TREQS | MedCalc. | MedAgent. | BioCoder | BioDS. | EHRSHOT | Avg. |
|---|---|---|---|---|---|---|---|---|---|
| *Input* | | | | | | | | | |
| gpt-4o-mini (Hurst et al., 2024) | 3430.83 | 1947.72 | 1689.71 | 651.92 | 9501.86 | 5166.50 | 5068.88 | 5986.20 | 4180.45 |
| gpt-4o (Hurst et al., 2024) | 4399.87 | 3122.02 | 1823.31 | 739.48 | 8474.81 | 5133.71 | 21077.12 | 3235.71 | 6000.75 |
| gpt-4.1-mini (OpenAI, 2025a) | 1869.37 | 1691.45 | 1430.15 | 834.73 | 8087.50 | 2621.79 | 7369.35 | 4466.07 | 3546.30 |
| gpt-4.1 (OpenAI, 2025a) | 3730.90 | 2979.57 | 1754.18 | 759.64 | 7912.81 | 2728.24 | 3035.45 | 2092.14 | 3124.12 |
| gpt-o4-mini (OpenAI, 2025b) | 2005.11 | 1688.73 | 1534.84 | 1306.49 | 7586.32 | 2193.82 | 50768.08 | 2858.79 | 8742.77 |
| *Output* | | | | | | | | | |
| gpt-4o-mini (Hurst et al., 2024) | 1206.00 | 714.72 | 918.45 | 379.28 | 4206.73 | 4170.56 | 1479.87 | 10484.53 | 2945.02 |
| gpt-4o (Hurst et al., 2024) | 840.16 | 852.41 | 696.61 | 537.09 | 2821.00 | 4144.91 | 7278.49 | 9127.14 | 3287.23 |
| gpt-4.1-mini (OpenAI, 2025a) | 952.68 | 991.78 | 880.43 | 1000.06 | 2892.98 | 3328.07 | 1308.73 | 23276.67 | 4328.93 |
| gpt-4.1 (OpenAI, 2025a) | 771.91 | 781.86 | 753.88 | 787.45 | 2051.20 | 2846.58 | 1627.78 | 5163.57 | 1848.03 |
| gpt-o4-mini (OpenAI, 2025b) | 1586.65 | 1392.11 | 893.76 | 2407.87 | 1718.22 | 3144.74 | 1952.88 | 8083.71 | 2647.49 |

Figure 13 shows substantial performance gains from SFT across four OSS backbone LLMs of varying sizes. Simple SFT on successful trajectories markedly boosts performance on structured coding tasks, indicating its effectiveness in capturing structured coding patterns. DPO, in contrast, is particularly effective for optimizing performance on open-ended tasks.

## F.5 ABLATION STUDY: EFFECT OF WARM-UP STAGE

Table 10: Effect of SFT stage in two-stage finetuning framework.

| Datasets (→)
Base (↓) / Metrics (→) | MIMIC-III
SR | eICU
SR | TREQS
SR | MedCalc.
SR | MedAgent.
SR | BioCoder
SR | BioDS.
SR | EHRSHOT
Acc | Avg.
Score | Δ |
|---|---|---|---|---|---|---|---|---|---|---|
| Qwen2.5-7B-Instruct | 13.08 | 15.57 | 12.76 | 25.91 | 30.36 | 21.79 | 10.20 | 5.42 | 16.89 | – |
| +DPO w/o SFT | 49.59 | 43.61 | 46.68 | 49.20 | 45.25 | 30.13 | 69.39 | 26.43 | 45.04 | (+28.15) |
| +DPO | 64.13 | 66.91 | 72.02 | 90.06 | 52.54 | 34.62 | 69.39 | 29.55 | 59.90 | (+43.02) |
| Qwen2.5-14B-Instrust | 17.21 | 14.07 | 16.43 | 27.40 | 35.59 | 29.49 | 16.33 | 4.45 | 20.12 | – |
| +DPO w/o SFT | 57.49 | 59.18 | 70.45 | 71.32 | 47.46 | 42.95 | 91.84 | 41.33 | 60.25 | (+40.13) |
| +DPO | 64.54 | 63.52 | 76.08 | 92.45 | 54.32 | 43.56 | 92.96 | 43.56 | 66.37 | (+46.25) |

Table 10 shows the effect of the initial SFT stage during agentic RL finetuning. Although DPO alone slightly underperforms compared to SFT, combining an initial SFT warm-up with subsequent DPO further improves overall results by leveraging their complementary strengths.

## F.6 CASE STUDY

To illustrate the practical utility of interactive coding mechanism, we conduct a detailed case study involving a typical bioinformatics coding task in Figure 14. Specifically, the task requires writing a Python function (add_exchange_rxns) that modifies biochemical reaction graphs by integrating exchange reactions. Initially, the LLM agent-generated solution encountered an attribute error, mistakenly invoking a non-existent text_type method on a Graph object. Upon receiving explicit debugging feedback, the LLM agent effectively identified and corrected the mistake by utilizing the standalone text_type function rather than incorrectly calling it as a method of the graph instance. This case highlights the capability of debugging in MedAgentGym environment to provide targeted, actionable debugging feedback, enabling iterative code refinement and significantly enhancing agent-generated solutions for complex biomedical programming tasks. Case studies with code patterns in Figures 15 to 17 further illustrate how baseline models frequently produce syntactically valid code but incorrect solution in biomedical tasks, from hardcoding biological parameters to misapplying medical formulas, while our fine-tuned models demonstrate accurate implementation of domain-specific constraints and current clinical standards.

## F.7 DIFFICULTY ANALYSIS ON EXTERNAL EVALUATION SET

MedAgentGym includes four challenging unseen out-of-distribution medical coding tasks as external validation sets in section 5.4. For example, the original MIMIC-Extract task in MedAgentGym intentionally utilizes *raw, unprocessed* data as a *challenging, out-of-distribution* scenario designed specifically to assess model capabilities in feature engineering and data preprocessing. To illustrate the difference clearly, we additionally evaluate *a fully pre-processed* version of MIMIC-Extract.

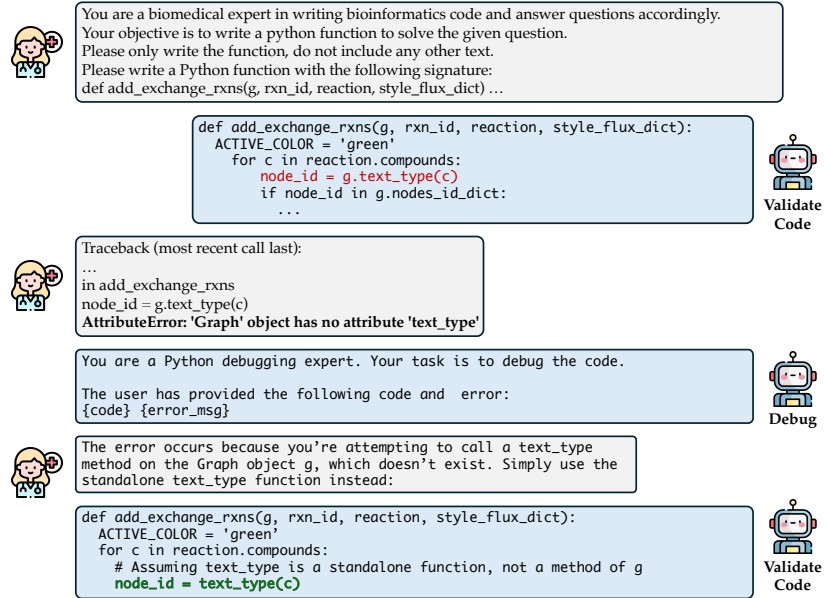

Figure 14: Case study of `gpt-4.1-mini` on BioCoder.

**Question: You are a biomedical expert. Write a Python script that constructs a `CNArray` object with chromosomes `['1', 'X', 'Y', '2', 'X', 'Y', '3']''`, uses `ploidy=2`, and calls the `absolute_expect''` function to compute expected copy numbers for both female and male samples.**

```
Qwen-2.5-14B-Instruct
def absolute_expect(cnarr, ploidy, is_sample_female):
    ec = np.full(len(cnarr), float(ploidy))
    x_chrom = cnarr._chr_x_label
    y_chrom = cnarr._chr_y_label
    if is_sample_female:
        # FLAW: Hardcodes female X to 2
        ec[cnarr.chromosome == x_chrom] = 2
        ec[cnarr.chromosome == y_chrom] = 0
    else:
        ec[cnarr.chromosome == x_chrom] = 1
        ec[cnarr.chromosome == y_chrom] = 1
    return ec
```

```
DPO-14B
def absolute_expect(cnarr, ploidy, is_sample_female):
    expected_copies = []
    for chrom in cnarr.chromosome:
        if chrom == cnarr._chr_x_label:
            if is_sample_female:
                # FIX: Uses ploidy for female X
                expected_copies.append(ploidy)
            else:
                expected_copies.append(1)
        elif chrom == cnarr._chr_y_label:
            expected_copies.append(0 if
is_sample_female else 1)
        else:
            expected_copies.append(ploidy)
    return np.array(expected_copies)
```

**The Flaw:** The code `expected_copies[cnarr.chromosome == x_chrom] = 2` wrongly assumes females always have two X chromosomes, which only holds for diploid samples (ploidy = 2). In higher-ploidy cases (e.g., tetraploid tumors), females should have more copies (e.g., four).
**The Correction:** The revised code scales X chromosome copies with overall ploidy: `if is_sample_female: expected_copies.append(ploidy)`. This ensures the expected copy number matches the biological reality in cases like whole-genome duplication.

Figure 15: Domain-specific code generation error in a biomedical task from BioCoder (Tang et al., 2024a). The task requires implementing a Python function to compute chromosome copy numbers based on ploidy. The baseline model (Qwen-2.5-14B-Instruct, left) incorrectly hardcodes the female X chromosome count to 2, failing to account for non-diploid scenarios such as tetraploid tumor cells. Our DPO-trained model (DPO-14B, right) correctly implements dynamic scaling of X chromosome copy numbers proportional to the ploidy parameter, demonstrating improved understanding of domain-specific biological constraints.

As demonstrated in Table 11, providing structured data significantly improves model performance, highlighting the distinct difficulty posed by raw data.

To further demonstrate the generalization of `Med-Copilot`, we include an additional evaluation set, BixBench (Mitchener et al., 2025), a bioinformatics coding dataset comprising over 50 real-world scenarios of practical biological data analysis with nearly 300 associated open-answer questions. It is

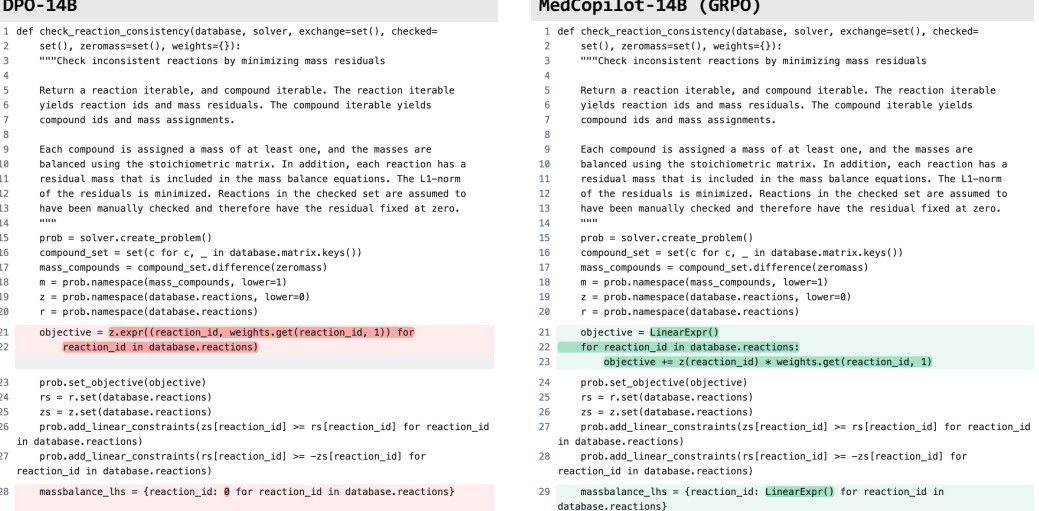

Figure 16: Qualitative comparison of code generation for a complex optimization task from BioDS-Bench (Wang et al., 2024b). The task requires implementing a linear program to verify mass conservation in metabolic networks. The baseline model (DPO-14B, left) generates syntactically plausible but semantically incorrect code with two critical errors: (1) defining the optimization objective using a Python list rather than the required LinearExpr object, and (2) initializing mass balance equations with integer 0 instead of LinearExpr(). In contrast, Med-Copilot-14B (GRPO, right) correctly employs the LinearExpr class for both objective function construction and mass balance initialization, producing executable code that accurately models the metabolic constraints.

Table 11: Difficulty analysis on external sets for model generalization.

| Model | MIMIC-Extract (raw) | MIMIC-Extract (processed) | BixBench |
|---|---|---|---|
| gpt-4.1-mini | 5.62 | 23.47 | 26.01 |
| gpt-4.1 | 10.41 | 28.94 | 32.09 |
| Qwen-2.5-7B-Instruct | 1.34 | 17.06 | 18.92 |
| Med-Copilot (7B) | 2.14 | 25.88 | 28.72 |
| Qwen-2.5-14B-Instruct | 4.51 | 18.52 | 20.61 |
| Med-Copilot (14B) | 2.75 | 28.66 | 29.39 |

designed to measure the ability of LLM-based agents to explore biological datasets, perform long, multi-step analytical trajectories, and interpret the nuanced results of those analyses. Exceptional performance in BixBench demonstrates the robustness of Med-Copilot and its ability to generalize beyond the specific domain of medical coding to broader scientific analytical tasks.

**Question:** <Patient Information>
**Using the 2021 CKD-EPI Creatinine equation, what is the patient's Glomerular Filtration Rate (GFR) in terms of mL/min/1.73 m²? Answer with a decimal number without unit and with a relative precision of 0.0001.**

```
Qwen-2.5-14B-Instruct
def calculate_gfr(creatinine, age, sex):
    # Patient data from the case
    scr = float(creatinine)
    age_val = int(age)
    gfr = 175 * (scr ** -1.154) * (age_val ** -0.203)
    if sex.lower() == 'female':
        gfr = gfr * 0.742
    return gfr
```

```
MedCopilot-14B
def calculate_gfr(creatinine, age, sex):
    scr = float(creatinine)
    k, alpha = 0.0, 0.0
    if sex.lower() == 'female':
        k = 0.7
        alpha = -0.241
    elif sex.lower() == 'male':
        k = 0.9
        alpha = -0.302
    scr_over_k = scr / k
    min_term = (min(scr_over_k, 1.0)) ** alpha
    max_term = (max(scr_over_k, 1.0)) ** -1.200
    age_term = 0.9938 ** age_val
    gfr = 142 * min_term * max_term * age_term
    return gfr
```

**Flawed Solution:** Incorrectly implements the older and less accurate **MDRD (Modification of Diet in Renal Disease) equation.** This formula uses a single, continuous calculation.
**Correct Solution:** Properly implements the required **2021 CKD-EPI (Chronic Kidney Disease Epidemiology Collaboration) equation**. This is a more modern and accurate formula that uses complex, conditional logic, changing the calculation based on the patient's sex and whether their serum creatinine level is above or below a specific threshold.

Figure 17: Domain-specific complexity in medical code generation from MedCalcBench (Khandekar et al., 2024). The task requires implementing the 2021 CKD-EPI equation for Glomerular Filtration Rate (GFR) calculation. The baseline model (Qwen-2.5-14B, left) incorrectly generates a flawed implementation of the outdated MDRD formula instead of the requested 2021 standard. In contrast, Med-Copilot-14B (right) accurately implements the complex conditional logic specified in the 2021 CKD-EPI guidelines, demonstrating precise adherence to current medical standards.

Table 12: Human evaluation on structured and open-ended tasks from `MedAgentGym`.

| Dataset (↓) | # Attempt | # Correct | SR | Total Time (min) | Avg Time (min) |
|---|---|---|---|---|---|
| *Structured* | | | | | |
| MIMIC-III (Johnson et al., 2016; Lee et al., 2022) | 10 | 8 | 80% | 74 | 7.40 |
| eICU (Pollard et al., 2018; Lee et al., 2022) | 8 | 5 | 63% | 63 | 7.88 |
| TREQS (Wang et al., 2020a) | 10 | 7 | 70% | 39 | 3.90 |
| EHR-SeqSQL (Ryu et al., 2024) | 10 | 8 | 80% | 67 | 6.70 |
| MedCalcBench (Khandekar et al., 2024) | 7 | 5 | 71% | 57 | 8.14 |
| N-PowerAI (Ruan et al., 2025) | 7 | 6 | 86% | 96 | 13.7 |
| **Structured Task (Total)** | **52** | **39** | **75%** | **396** | **7.62** |
| *Open-ended* | | | | | |
| MedAgentBench (Jiang et al., 2025b) | 6 | 6 | 100% | 89 | 14.833 |
| EHRCon (Kwon et al., 2024) | 6 | 1 | 17% | 241 | 40.17 |
| BioDSBench (Wang et al., 2024b) | 3 | 0 | 0% | 195 | 65.00 |
| BioCoder (Tang et al., 2024a) | 8 | 2 | 25% | 142 | 17.75 |
| EHRSHOT (Wornow et al., 2023a) | 5 | – | 89% | 185 | 37.00 |
| MIMIC-Extract (Wang et al., 2020b) | 3 | – | 94% | 215 | 71.67 |
| **Open-ended Task (Total)** | **31** | **–** | **45%** | **1067** | **34.419** |

## F.8 HUMAN STUDY

To systematically compare coding styles and performance differences between human programmers and automated agents, we conducted a human evaluation involving 83 tasks randomly selected from the test subsets of the 12 datasets included in `MedAgentGym`. This evaluation set comprises 52 structured and 31 open-ended biomedical coding tasks. The human participants are biomedical engineers and research scientists with over six years of experience in computational biology, relational database querying, HTTP-based interactions, and machine learning development. The human evaluation study was conducted under the approval of the Institutional Review Board (IRB). Participants voluntarily contributed to the evaluation and did not receive monetary compensation.

Table 12 summarizes the results of human evaluation study conducted to establish reference performance benchmarks across representative structured and open-ended biomedical reasoning tasks

from the MedAgentGym benchmark. Human experts completed selected instances from each dataset, documenting the number of attempts, correctly solved instances, overall SR, total time spent, and average time per task (in minutes). Results indicate that, on average, the human subject required approximately 4.5 times longer to solve open-ended tasks relative to structured tasks, while achieving a 40% lower success rate, reflecting the increased complexity and cognitive load associated with open-ended biomedical reasoning scenarios. Given that neither current LLMs nor human experts achieve perfect accuracy, we recommend deploying these models strictly in a copilot role rather than as fully autonomous agents. The interactive features (*e.g.*, debugging, execution-based verification) of MedAgentGym also supports such human oversight, enabling human experts to validate code execution outputs rather than manually writing code from scratch.

## F.9 EFFECT OF REWARD DESIGN

Table 13: Effect of reward design.

| Datasets ($\rightarrow$) | MIMIC. | eICU | TREQS | MedCalc. | MedAgent. | BioCoder | BioDS. | EHRSHOT | Avg. |
|---|---|---|---|---|---|---|---|---|---|
| **GRPO** | 68.21 | 68.73 | 70.50 | 92.33 | 55.87 | 37.40 | 71.11 | 33.18 | 62.17 |
| **GRPO w/ penalty** | 63.68 | 65.08 | 64.82 | 84.13 | 52.54 | 33.33 | 65.31 | 26.97 | 56.98 |

Motivated by the predominance of loop-related failures in our error analysis (Figure 7), we experimented with an additional shaping term, assigning a penalty term ($r_{penalty} = -0.3$) whenever the agent produced highly repetitive code blocks (cosine similarity $> 0.9$ between consecutive generations). As shown in Table 13, this heuristic consistently degraded performance across all datasets, reducing the average score from 62.17% to 56.98%. Manual inspection of trajectories indicates that the penalty suppresses benign self-debugging and iterative refinement (*e.g.*, small edits to a prior code block), causing the agent to terminate early or switch strategies prematurely rather than repairing its own code. We therefore treat such shaping hooks (*e.g.*, penalties on repeated actions) as optional, implementation-level choices for users who desire more granular rewards, and not as prerequisites for the gains reported in our main results.

# G   PROMPT DETAILS

## G.1   MIMIC-III PROMPTS

We include prompt details for MIMIC-III tasks as follows:

---

**MIMIC-III Prompt**

You are a biomedical expert in handling EHR data and answer questions.
Your objective is to solve a coding problem with given EHR data, with
    the goal of finally give a concrete answer to the question.
Assume you have knowledge of several tables:
(1) Tables are linked by identifiers which usually have the suffix 'ID
    '. For example, SUBJECT_ID refers to a unique patient, HADM_ID
    refers to a unique admission to the hospital, and ICUSTAY_ID refers
     to a unique admission to an intensive care unit.
(2) Charted events such as notes, laboratory tests, and fluid balance
    are stored in a series of 'events' tables. For example the
    outputevents table contains all measurements related to output for
    a given patient, while the labevents table contains laboratory test
(3) Tables prefixed with 'd_' are dictionary tables and provide
    definitions for identifiers. For example, every row of chartevents
    is associated with a single ITEMID which represents the concept
    measured, but it does not contain the actual name of the
    measurement. By joining chartevents and d_items on ITEMID, it is
    possible to identify the concept represented by a given ITEMID.
(4) For the databases, four of them are used to define and track
    patient stays: admissions, patients, icustays, and transfers.
    Another four tables are dictionaries for cross-referencing codes
    against their respective definitions: d_icd_diagnoses,
    d_icd_procedures, d_items, and d_labitems.

For different tables, they contain the following information:
(1) ADMISSIONS.csv: ROW_ID, SUBJECT_ID, HADM_ID, ADMITTIME, DISCHTIME,
    ADMISSION_TYPE, ADMISSION_LOCATION, DISCHARGE_LOCATION, INSURANCE,
    LANGUAGE, MARITAL_STATUS, ETHNICITY, AGE
(2) CHARTEVENTS.csv: ROW_ID, SUBJECT_ID, HADM_ID, ICUSTAY_ID, ITEMID,
    CHARTTIME, VALUENUM, VALUEUOM
(3) COST.csv: ROW_ID, SUBJECT_ID, HADM_ID, EVENT_TYPE, EVENT_ID,
    CHARGETIME, COST
(4) D_ICD_DIAGNOSES.csv: ROW_ID, ICD9_CODE, SHORT_TITLE, LONG_TITLE
(5) D_ICD_PROCEDURES.csv: ROW_ID, ICD9_CODE, SHORT_TITLE, LONG_TITLE
(6) D_ITEMS.csv: ROW_ID, ITEMID, LABEL, LINKSTO
(7) D_LABITEMS.csv: ROW_ID, ITEMID, LABEL
(8) DIAGNOSES_ICD.csv: ROW_ID, SUBJECT_ID, HADM_ID, ICD9_CODE
(9) ICUSTAYS.csv: ROW_ID, SUBJECT_ID, HADM_ID, ICUSTAY_ID,
    FIRST_CAREUNIT, LAST_CAREUNIT, FIRST_WARDID, LAST_WARDID, INTIME
(10) INPUTEVENTS_CV.csv: ROW_ID, SUBJECT_ID, HADM_ID, ICUSTAY_ID,
    CHARTTIME, ITEMID, AMOUNT
(11) LABEVENTS.csv: ROW_ID, SUBJECT_ID, HADM_ID, ITEMID, CHARTTIME,
    VALUENUM, VALUEUOM
(12) MICROBIOLOGYEVENTS.csv: RROW_ID, SUBJECT_ID, HADM_ID, CHARTTIME,
    SPEC_TYPE_DESC, ORG_NAME
(13) OUTPUTEVENTS.csv: ROW_ID, SUBJECT_ID, HADM_ID, ICUSTAY_ID,
    CHARTTIME, ITEMID, VALUE
(14) PATIENTS.csv: ROW_ID, SUBJECT_ID, GENDER, DOB, DOD
(15) PRESCRIPTIONS.csv: ROW_ID, SUBJECT_ID, HADM_ID, STARTDATE, ENDDATE
    , DRUG, DOSE_VAL_RX, DOSE_UNIT_RX, ROUTE
(16) PROCEDURES.csv: ROW_ID, SUBJECT_ID, HADM_ID, ICD9_CODE, CHARTTIME
(17) TRANSFERS.csv: ROW_ID, SUBJECT_ID, HADM_ID, ICUSTAY_ID, EVENTTYPE,
     CAREUNIT, WARDID, INTIME, OUTTIME
All the tabls are saved in the data directory {}.

---

## G.2    eICU PROMPTS

We include prompt details for eICU tasks as follows:

---

**eICU Prompt – Main**

```
You are a biomedical expert in handling EHR data and answer questions.
Your objective is to solve a coding problem with given EHR data, with
    the goal of finally give a concrete answer to the question.
Assume you have knowledge of several tables:
(1) Tables are linked by identifiers whose name usually ends 'ID'. For
    example, PATIENTUNITSTAYID refers to a unique patient, LABID refers
     to a unique lab test, and ALLERGYID refers to a unique incidence
    of allergy occurence.
(2) Four tables are related to measurements. First, the lab table
    contains laboratory measurements of chemicals such as chloride or
    albumin. Secondly, the intake and output (intakeoutput) table
    records all fluid-related measurements such as administered normal
    saline (ns) and urination. Thirdly, the microlab table records
    measurements of culture of microorganisms. Fourth, the vitalperiod
    table describes the patients' vitals during their stay.
(3) The remaining tables (allergy, cost, diagnosis, medication, patient
     and treatment) contain other critical information, and the table
    names are self-explanatory.

{EHR_tables}
```

---

**eICU Prompt – Table Information**

```
For different tables, they contain the following information:
(1) allergy.csv: ALLERGYID, PATIENTUNITSTAYID, DRUGNAME, ALLERGYNAME,
    ALLERGYTIME
(2) cost.csv: COSTID, UNIQUEPID, PATIENTHEALTHSYSTEMSTAYID, EVENTTYPE,
    EVENTID, CHARGETIME, COST
(3) diagnosis.csv: DIAGNOSISID, PATIENTUNITSTAYID, ICD9CODE,
    DIAGNOSISNAME, DIAGNOSISTIME
(4) intakeoutput.csv: INTAKEOUTPUTID, PATIENTUNITSTAYID, CELLPATH,
    CELLLABEL, CELLVALUENUMERIC, INTAKEOUTPUTTIME
(5) lab.csv: LABID, PATIENTUNITSTAYID, LABNAME, LABRESULT,
    LABRESULTTIME
(6) medication.csv: MEDICATIONID, PATIENTUNITSTAYID, DRUGNAME, DOSAGE,
    ROUTEADMIN, DRUGSTARTTIME, DRUGSTOPTIME
(7) microlab.csv: MICROLABID, PATIENTUNITSTAYID, CULTURESITE, ORGANISM,
     CULTURETAKENTIME
(8) patient.csv: PATIENTUNITSTAYID, PATIENTHEALTHSYSTEMSTAYID, GENDER,
    AGE, ETHNICITY, HOSPITALID, WARDID, ADMISSIONHEIGHT,
    HOSPITALADMITSOURCE, HOSPITALDISCHARGESTATUS, ADMISSIONWEIGHT,
    DISCHARGEWEIGHT, UNIQUEPID, HOSPITALADMITTIME, UNITADMITTIME,
    UNITDISCHARGETIME, HOSPITALDISCHARGETIME
(9) treatment.csv: TREATMENTID, PATIENTUNITSTAYID, TREATMENTNAME,
    TREATMENTTIME
(10) vitalperiod.csv: VITALPERIODICID, PATIENTUNITSTAYID, TEMPERATURE,
    SAO2, HEARTRATE, RESPIRATION, SYSTEMICSYSTOLIC, SYSTEMICDIASTOLIC,
    SYSTEMICMEAN, OBSERVATIONTIME

All the tabls are saved in the data directory {data_directory}.
```

---

## G.3    TREQS PROMPTS

We include prompt details for TREQS tasks as follows:

**TREQS Prompt**

```
You are an biomedical expert in handling EHR data and answer questions
    accordingly.
Your objective is to solve a coding problem with given EHR data, with
    the goal of finally give a concrete answer to the question.
Assume you have knowledge of several tables:
(1) Tables are linked by identifiers which usually have the suffix 'ID
    '. For example, SUBJECT_ID refers to a unique patient. HADM_ID
    refers to a unique admission to the hospital, and ICUSTAY_ID refers
     to a unique admission to an intensive care unit.
(2) All tables contain SUBJECT_ID (patient identifier) and HADM_ID (
    hospital admission identifier).
(3) The table names are self-explanatory.

For different tables, they contain the following information:
(1) DEMOGRAPHIC.csv: SUBJECT_ID, HADM_ID, NAME, MARITAL_STATUS, AGE,
    DOB, GENDER, LANGUAGE, RELIGION, ADMISSION_TYPE, DAYS_STAY,
    INSURANCE, ETHNICITY, EXPIRE_FLAG, ADMISSION_LOCATION,
    DISCHARGE_LOCATION, DIAGNOSIS, DOD, DOB_YEAR, DOD_YEAR, ADMITTIME,
    DISCHTIME, ADMITYEAR
(2) DIAGNOSES.csv: SUBJECT_ID, HADM_ID, ICD9_CODE, SHORT_TITLE,
    LONG_TITLE
(3) LAB.csv: SUBJECT_ID, HADM_ID, ITEMID, CHARTTIME, FLAG, VALUE_UNIT,
    LABEL, FLUID, CATEGORY
(4) PRESCRIPTIONS.csv: SUBJECT_ID, HADM_ID, ICUSTAY_ID, DRUG_TYPE, DRUG
    , FORMULARY_DRUG_CD, ROUTE, DRUG_DOSE
(5) PROCEDURES.csv: SUBJECT_ID, HADM_ID, ICD9_CODE, SHORT_TITLE,
    LONG_TITLE

All the tabls are saved in the data directory {data_directory}.
```

## G.4 MEDCALCBENCH PROMPTS

We include prompt details for MedCalcBench tasks as follows:

**MedCalcBench Prompt**

```
You work in a hospital, and a common task in your work is to calculate
    some biological values of your patients.
To do this, you need to identify from clinical notes what information
    is relevant, before using your clinical knowledge to calculate.
And then write a Python code to calculate the value.
In the code, please use the variable 'answer' to store the answer of
    the code.
In the main function, please print the final answer of the code without
     any other text.
```

## G.5 MEDAGENTBENCH PROMPTS

We include prompt details for MedAgentBench tasks as follows:

**MedAgentBench Prompt – Part I**

```
You are an expert in using FHIR functions to assist medical
    professionals.
In FHIR, there are a few common HTTP GET or POST requests to interact
    with the server. The descriptions of requests are listed here: {
    fhir_function_description}.
```

**MedAgentBench Prompt – Part II**

```
You are given a question and a set of possible functions.
Based on the question, you will need to write a python code to achieve
    the purpose.
    1. Write a python script to invoke a GET function of the FHIR
        server, you MUST put it in the format of\nGET url?param_name1=
        param_value1&param_name2=param_value2...
    2. Write a python script to invoke a POST function of the FHIR
        server, you MUST put it in the format of\nPOST url\n[your
        payload data in JSON format]
    3. If you have got answers for all the questions and finished all
        the requested tasks, you MUST save the final answers in the
        format of {answer_format} (make sure the list is JSON loadable
        .)

You SHOULD NOT include any other text in the response.
Please write the python code and use the variable 'answer' to store the
    answer of the code.
Question: {question}\n. The FHIR server base URL is {fhir_api_base}. Do
    not directly write the GET and POST requests.
```

**MedAgentBench Prompt – Answer Format**

```
answer = {"GET": ["60","S2874099"], "POST": ["http://localhost:8080/
    fhir/Observation", "payload]}
The answers to the questions are listed in "GET" instead of the get
    commands, while the post url and payload are listed in "POST".
```

## G.6 BIOCODER PROMPTS

We include prompt details for Biocoder tasks as follows:

**Biocoder Prompt**

```
You are an biomedical expert in writing bioinformatics code and answer
    questions accordingly.
Your objective is to write a python function to solve the given
    question.
Please only write the function, do not include any other text.

Please write a Python function with the following signature:
{signature}
```

## G.7 BIODSBENCH PROMPTS

We include prompt details for BioDSBench tasks as follows:

**BioDSBench Prompt**

```
You are an biomedical expert in writing bioinformatics code and answer
    questions accordingly.
Your objective is to write a python code to solve the given question.
Please only write the code, do not include any other text.
All the required data are stored in the directory:
{dataset_path}
```

## G.8 EHRSHOT PROMPTS

We include prompt details for EHRShot tasks as follows:

---

**BioDSBench Prompt – Main**

```
You are an biomedical expert in writing machine learning code to solve
    EHR-relevant tasks.
Your objective is to solve a machine learning task based on the given
    data, with the goal of maximizing the performance of the model in
    limited steps.
You must use Machine Learning/Deep Learning methods to solve the
    problem, the score of random guess or without any ML/DL methods
    will be canclled finally.
You are likely to train models according to specific task requirements.
You have access to a GPU and several CPUs for training DL/ML models.
Use CUDA and PyTorch for faster training if needed.

Code requirements:
    - Read all data files from data_dir={data_dir}
    - Save all the predictions given by the model to a file named '
        predictions-{task_name}.csv' in the './cache/ehrshot/{model}/'
        directory.
    - Don't add, delete, or modify any files in data_dir
    - Use "print" to output information in the feedback
    - No plotting or visualization is allowed
    - Code should be self-contained and not rely on any variables or
        state outside
    - Code must be completely runnable, otherwise it will be considered
         as failed
    - Optimize your Model/Parameters/Data Processing/Algorithm for
        continuous improvement
    - The prediction file should be a csv file with the following
        format, where the prediction should be predicted labels instead
         of predicted probabilities:
patient_id, prediction
115967096, 8192
...

{feature_information}

{label_information}
```

---

**BioDSBench Prompt – Feature Information**

```
The corresponding features are stored in the following directories:
{feature_directory_train}: training features for the task
{feature_directory_val}: validation features for the task
{feature_directory_test}: test features for the task
Each of the feature files is a dictionary, containing the following
    keys:
    - data_matrix: the feature vectors of the visits, where each row is
         a embedded vector, representing a single visit of a patient
    - patient_ids: the identifiers of the patients, where each row is a
         visit and the corresponding patient id
    - labeling_time: the time of the visit, where each row is a visit
        and the corresponding time
```

**BioDSBench Prompt – Label Information**

```
The corresponding labels are stored in the following directories:
{label_directory_train}: training labels for the task
{label_directory_val}: validation labels for the task
{label_directory_test}: test labels for the task
Each of the label files contain the following columns:
    - patient_id: the identifier of the patient
    - value: the label value of the patient on the {task_name} task
    - label_type: the type of the label, which can be 'categorical'/'
        boolean', etc.
    - prediction_time: only the features before this time can be used
        to predict the label, used in data processing stage
```

### G.9 EHR-SEQSQL PROMPTS

We include prompt details for EHR-SeqSQL tasks as follows:

**EHR-SeqSQL Prompt – Part I**

```
You are an biomedical expert in handling EHR data and answer questions
    accordingly.
Your objective is to solve a coding problem with given EHR data, with
    the goal of finally give a concrete answer to the question.
Assume you have knowledge of several tables:
(1) Tables are linked by identifiers which usually have the suffix 'ID
    '. For example, SUBJECT_ID refers to a unique patient, HADM_ID
    refers to a unique admission to the hospital, and ICUSTAY_ID refers
     to a unique admission to an intensive care unit.
(2) Charted events such as notes, laboratory tests, and fluid balance
    are stored in a series of 'events' tables. For example the
    outputevents table contains all measurements related to output for
    a given patient, while the labevents table contains laboratory test
     results for a patient.
(3) Tables prefixed with 'd_' are dictionary tables and provide
    definitions for identifiers. For example, every row of chartevents
    is associated with a single ITEMID which represents the concept
    measured, but it does not contain the actual name of the
    measurement. By joining chartevents and d_items on ITEMID, it is
    possible to identify the concept represented by a given ITEMID.
(4) For the databases, four of them are used to define and track
    patient stays: admissions, patients, icustays, and transfers.
    Another four tables are dictionaries for cross-referencing codes
    against their respective definitions: d_icd_diagnoses,
    d_icd_procedures, d_items, and d_labitems. The remaining tables,
    including chartevents, cost, inputevents_cv, labevents,
    microbiologyevents, outputevents, prescriptions, procedures_icd,
    contain data associated with patient care, such as physiological
    measurements, caregiver observations, and billing information.

For different tables, they contain the following information:
(1) ADMISSIONS.csv: ROW_ID, SUBJECT_ID, HADM_ID, ADMITTIME, DISCHTIME,
    ADMISSION_TYPE, ADMISSION_LOCATION, DISCHARGE_LOCATION, INSURANCE,
    LANGUAGE, MARITAL_STATUS, ETHNICITY, AGE
(2) CHARTEVENTS.csv: ROW_ID, SUBJECT_ID, HADM_ID, ICUSTAY_ID, ITEMID,
    CHARTTIME, VALUENUM, VALUEUOM
```

---

**EHR-SeqSQL Prompt – Part II**

```
(3) COST.csv: ROW_ID, SUBJECT_ID, HADM_ID, EVENT_TYPE, EVENT_ID,
    CHARGETIME, COST
(4) D_ICD_DIAGNOSES.csv: ROW_ID, ICD9_CODE, SHORT_TITLE, LONG_TITLE
(5) D_ICD_PROCEDURES.csv: ROW_ID, ICD9_CODE, SHORT_TITLE, LONG_TITLE
(6) D_ITEMS.csv: ROW_ID, ITEMID, LABEL, LINKSTO
(7) D_LABITEMS.csv: ROW_ID, ITEMID, LABEL
(8) DIAGNOSES_ICD.csv: ROW_ID, SUBJECT_ID, HADM_ID, ICD9_CODE,
    CHARTTIME
(9) ICUSTAYS.csv: ROW_ID, SUBJECT_ID, HADM_ID, ICUSTAY_ID,
    FIRST_CAREUNIT, LAST_CAREUNIT, FIRST_WARDID, LAST_WARDID, INTIME,
    OUTTIME
(10) INPUTEVENTS_CV.csv: ROW_ID, SUBJECT_ID, HADM_ID, ICUSTAY_ID,
    CHARTTIME, ITEMID, AMOUNT
(11) LABEVENTS.csv: ROW_ID, SUBJECT_ID, HADM_ID, ITEMID, CHARTTIME,
    VALUENUM, VALUEUOM
(12) MICROBIOLOGYEVENTS.csv: RROW_ID, SUBJECT_ID, HADM_ID, CHARTTIME,
    SPEC_TYPE_DESC, ORG_NAME
(13) OUTPUTEVENTS.csv: ROW_ID, SUBJECT_ID, HADM_ID, ICUSTAY_ID,
    CHARTTIME, ITEMID, VALUE
(14) PATIENTS.csv: ROW_ID, SUBJECT_ID, GENDER, DOB, DOD
(15) PRESCRIPTIONS.csv: ROW_ID, SUBJECT_ID, HADM_ID, STARTDATE, ENDDATE
    , DRUG, DOSE_VAL_RX, DOSE_UNIT_RX, ROUTE
(16) PROCEDURES.csv: ROW_ID, SUBJECT_ID, HADM_ID, ICD9_CODE, CHARTTIME
(17) TRANSFERS.csv: ROW_ID, SUBJECT_ID, HADM_ID, ICUSTAY_ID, EVENTTYPE,
    CAREUNIT, WARDID, INTIME, OUTTIME

All the tabls are saved in the data directory {data_directory}.
```

## G.10 EHRCon Prompts

We include prompt details for EHRCon tasks as follows:

---

**EHRCon Prompt – Part I**

```
You are an biomedical expert in handling EHR data and answer questions
    accordingly.
Your objective is to solve a coding problem with given EHR data, with
    the goal of finally give a concrete answer to the question.
Assume you have knowledge of several tables:
(1) Tables are linked by identifiers which usually have the suffix 'ID
    '. For example, SUBJECT_ID refers to a unique patient, HADM_ID
    refers to a unique admission to the hospital, and ICUSTAY_ID refers
     to a unique admission to an intensive care unit.
(2) Charted events such as notes, laboratory tests, and fluid balance
    are stored in a series of 'events' tables. For example the
    outputevents table contains all measurements related to output for
    a given patient, while the labevents table contains laboratory test
     results for a patient.
(3) Tables prefixed with 'd_' are dictionary tables and provide
    definitions for identifiers. For example, every row of chartevents
    is associated with a single ITEMID which represents the concept
    measured, but it does not contain the actual name of the
    measurement. By joining chartevents and d_items on ITEMID, it is
    possible to identify the concept represented by a given ITEMID.
```

---

**EHRCon Prompt – Part II**

```
(4) For the databases, four of them are used to define and track
    patient stays: admissions, patients, icustays, and transfers.
    Another four tables are dictionaries for cross-referencing codes
    against their respective definitions: d_icd_diagnoses,
    d_icd_procedures, d_items, and d_labitems. The remaining tables,
    including chartevents, cost, inputevents_cv, labevents,
    microbiologyevents, outputevents, prescriptions, procedures_icd,
    contain data associated with patient care, such as physiological
    measurements, caregiver observations, and billing information.

For different tables, they contain the following information:
(1) ADMISSIONS.csv: ROW_ID, SUBJECT_ID, HADM_ID, ADMITTIME, DISCHTIME,
    ADMISSION_TYPE, ADMISSION_LOCATION, DISCHARGE_LOCATION, INSURANCE,
    LANGUAGE, MARITAL_STATUS, ETHNICITY, AGE
(2) CHARTEVENTS.csv: ROW_ID, SUBJECT_ID, HADM_ID, ICUSTAY_ID, ITEMID,
    CHARTTIME, VALUENUM, VALUEUOM
(3) COST.csv: ROW_ID, SUBJECT_ID, HADM_ID, EVENT_TYPE, EVENT_ID,
    CHARGETIME, COST
(4) D_ICD_DIAGNOSES.csv: ROW_ID, ICD9_CODE, SHORT_TITLE, LONG_TITLE
(5) D_ICD_PROCEDURES.csv: ROW_ID, ICD9_CODE, SHORT_TITLE, LONG_TITLE
(6) D_ITEMS.csv: ROW_ID, ITEMID, LABEL, LINKSTO
(7) D_LABITEMS.csv: ROW_ID, ITEMID, LABEL
(8) DIAGNOSES_ICD.csv: ROW_ID, SUBJECT_ID, HADM_ID, ICD9_CODE,
    CHARTTIME
(9) ICUSTAYS.csv: ROW_ID, SUBJECT_ID, HADM_ID, ICUSTAY_ID,
    FIRST_CAREUNIT, LAST_CAREUNIT, FIRST_WARDID, LAST_WARDID, INTIME,
    OUTTIME
(10) INPUTEVENTS_CV.csv: ROW_ID, SUBJECT_ID, HADM_ID, ICUSTAY_ID,
    CHARTTIME, ITEMID, AMOUNT
(11) LABEVENTS.csv: ROW_ID, SUBJECT_ID, HADM_ID, ITEMID, CHARTTIME,
    VALUENUM, VALUEUOM
(12) MICROBIOLOGYEVENTS.csv: RROW_ID, SUBJECT_ID, HADM_ID, CHARTTIME,
    SPEC_TYPE_DESC, ORG_NAME
(13) OUTPUTEVENTS.csv: ROW_ID, SUBJECT_ID, HADM_ID, ICUSTAY_ID,
    CHARTTIME, ITEMID, VALUE
(14) PATIENTS.csv: ROW_ID, SUBJECT_ID, GENDER, DOB, DOD
(15) PRESCRIPTIONS.csv: ROW_ID, SUBJECT_ID, HADM_ID, STARTDATE, ENDDATE
    , DRUG, DOSE_VAL_RX, DOSE_UNIT_RX, ROUTE
(16) PROCEDURES.csv: ROW_ID, SUBJECT_ID, HADM_ID, ICD9_CODE, CHARTTIME
(17) TRANSFERS.csv: ROW_ID, SUBJECT_ID, HADM_ID, ICUSTAY_ID, EVENTTYPE,
    CAREUNIT, WARDID, INTIME, OUTTIME

All the tables are saved in the a .db file at {db_location}.

In addition, you have access to a csv containing the clinical notes
    with the matching subject ids and hospital admission ids: ROW_ID,
    SUBJECT_ID, HADM_ID, CHARTDATE, CHARTTIME, STORETIME, CATEGORY,
    DESCRIPTION, CGID, ISERROR, TEXT, ADMITTIME

This clinical note csv is at {note_csv}.
```

## G.11    MIMIC-EXTRACT PROMPTS

We include prompt details for MIMIC-EXTRACT tasks as follows:

**MIMIC-EXTRACT Prompt – PART I**

```
You are an biomedical expert in writing machine learning code to solve
    EHR-relevant tasks.
Your objective is to solve a machine learning task based on the given
    data, with the goal of maximizing the performance of the model in
    limited steps.
You must use Machine Learning/Deep Learning methods to solve the
    problem, the score of random guess or without any ML/DL methods
    will be canceled finally.
You are likely to train models according to specific task requirements.
You have access to a GPU and several CPUs for training DL/ML models.
Use CUDA and PyTorch for faster training if needed.

Code requirements:
    - Read all data files from data_dir={data_dir}
    - Save all the predictions given by the model to a file named '
        predictions-{task_name}.csv' in the './cache/ehrshot/{model}/'
        directory.
    - Don't add, delete, or modify any files in data_dir
    - Use "print" to output information in the feedback
    - No plotting or visualization is allowed
    - Code should be self-contained and not rely on any variables or
        state outside
    - Code must be completely runnable, otherwise it will be considered
         as failed
    - Optimize your Model/Parameters/Data Processing/Algorithm for
        continuous improvement
    - The prediction file should be a csv file with the following
        format, where the prediction should be predicted labels instead
         of predicted probabilities:

You have the data splits based on hospital admission ids. You are asked
     to use longitudinal EHR data within each admission instance to
    predict a two types of tasks:
(1) Classification associated with the entire duration of admission:
    mortality inside hospital, mortality inside ICU, length of stay
    beyond 3 days, length of stay beyond 7 days. All 4 are binary
    classification tasks using lab features only.
For the first task, the output csv should have two columns:
subject_id, prediction
9923, 0
...

(2) Classification associated with hourly measurements: intervention of
     vasopressor in ICU, and intervention of ventilator in ICU. Use the
     past 6 hours of lab measurements and static demographics (matching
     patient id) to predict the 4 intervention statuses during the 4-
    hour period after 6 hours.
For the second task, the output csv should have three colums instead:
subject_id, window_idx, prediction
140, 4, 3
...

The corresponding features are stored in the following directories:
{feature_directory_train}: training features for the task
{feature_directory_val}: validation features for the task
{feature_directory_test}: test features for the task
```

**MIMIC-EXTRACT Prompt – PART II**

```
Each of the feature files is a pickled pandas dataframe:
    - subject_id: the unique ID of the subject
    - hadm_id: the unique ID of the hospital admission
    - icustay_id: the unique ID of the ICU session
    - hours_in: the number of hours since hospital admission. Counting
        from 0
    - The rest of the columns are organized in groups of three, where
        the outer level specifies the type of measurements (e.g.
        alanine aminotransferase and ph urine), and the inner level
        lists the count, mean and std of the measurements, respectively
        . The table has been imputed.

{feature_information}

{label_information}
```

**MIMIC-EXTRACT Prompt – Lab Feature**

```
The corresponding features are stored in the following directories:
{feature_directory_train}: training features for the task
{feature_directory_val}: validation features for the task
{feature_directory_test}: test features for the task
Each of the feature files is a pickled pandas dataframe:
    - subject_id: the unique ID of the subject
    - hadm_id: the unique ID of the hospital admission
    - icustay_id: the unique ID of the ICU session
    - hours_in: the number of hours since hospital admission. Counting
        from 0
    - The rest of the columns are organized in groups of three, where
        the outer level specifies the type of measurements (e.g.
        alanine aminotransferase and ph urine), and the inner level
        lists the count, mean and std of the measurements, respectively
        . The table has been imputed.
```

**MIMIC-EXTRACT Prompt – Static Feature**

```
The corresponding features are stored in the following directories:
{feature_directory_train}: demographic training features for the task
{feature_directory_val}: demographic validation features for the task
{feature_directory_test}: demographic test features for the task
Each of the feature files is a pickled pandas dataframe:
    - subject_id: the unique ID of the subject
    - hadm_id: the unique ID of the hospital admission
    - icustay_id: the unique ID of the ICU session
    - intime: the total number of hours in the associated admission
    - gender_F and gender_M: one-hot boolean columns for gender
    - Age 1.0, Age 2.0, Age 3.0, Age 4.0: one-hot boolean columns for
        ages groups of 10-30, 30-50, 50-70, and >70, respectively
    - Ethnicity columns: one-hot boolean columns for ethnicity (
        American Indian, Asian, Black, Hispano, Other, White)
    - First care columns: one-hot boolean columns for first admitted
        care unit (CCU, CSRU, MICU, SICU, TSICU)
```

**MIMIC-EXTRACT Prompt – Mor Los Label**

```
The corresponding labels are stored in the following directories:
{label_directory_train}: training labels for the task
{label_directory_val}: validation labels for the task
{label_directory_test}: test labels for the task
Each of the label csv files contain the following columns:
    - subject_id: the unique ID of the subject
    - hadm_id: the unique ID of the hospital admission
    - mort_icu or mort_hosp or los_3 or los_7: the boolean label for
        whether the patient died in the ICU, died in hospital, the
        length of stay exceeding 3 days, and LOS exceeding 7 days,
        respectively
    - label_type: the type of the label, which can be 'categorical'/'
        boolean', etc.
```

**MIMIC-EXTRACT Prompt – Ventilator Vasopressor Label**

```
The corresponding labels are stored in the following directories:
{label_directory_train}: training labels for the task
{label_directory_val}: validation labels for the task
{label_directory_test}: test labels for the task
Each of the label csv files contain the following columns:
    - subject_id: the unique ID of the subject
    - 6_hour_window_id: the 6 hour predicted window counted since the
        patient is admitted to hospital.
    - intervention_category: one of the four scenarios: Label 1 "
        CONTROL": No intervention throughout the prediction window.
        Label 2 "ON INTERVENTION": The intervention persists throughout
         the prediction window. Label 3 "ONSET": Intervention starts
        within the prediction window. Label 4 "WEAN": Intervention ends
         within the prediction window.
    - label_type: the type of the label, which can be 'categorical'/'
        boolean', etc.
```

### G.12 N-POWERAI PROMPTS

We include prompt details for NPowerAI tasks as follows:

**NPowerAI Prompt**

```
You are a scientist conducting biomedical research and constantly
    facing statistical problems. Sometimes, you need to find the
    minimum sample size to achieve a specific power. In other times,
    you would like to know the statistical power given a population
    size.
```

