# OpenReview forum: "MedAgentGym: A Scalable Agentic Training Environment for Code-Centric Reasoning in Biomedical Data Science"
_ICLR.cc/2026/Conference — ICLR 2026 Oral_

### Official Review · Reviewer_Keuw · 2025-10-22

**Soundness:** 3
**Presentation:** 3
**Contribution:** 3
**Rating:** 6
**Confidence:** 3

**Summary:**

This paper introduced MedAgentGym, a training environment for coding-based biomedical agents. It consists of three folds of contributions: (1) MedAgentGym involves 72,413 task instances across 129 categories derived from 12 biomedical scenarios. (2) This training platform allows for efficient deployment and scalable evaluation, benchmarking 29 LLMs. (3) The training data collected from MedAgentBench leads to the powerful Med-Copilot-7B/14B, which produce comparable results as the much larger proprietary LLMs.

**Strengths:**

S1: The preparation of MedAgentGym requires a significant amount of effort, and is a non-trivial contribution to the open-source community.
S2: The setup in MedAgentGym is comprehensive, and the evaluation of existing LLMs on coding-based medical reasoning tasks is thorough.
S3: Med-Copilot-7B and -14B are also very useful open-source models for medical reasoning tasks, and their training and testing setups are solid.

**Weaknesses:**

W1: How are the tasks studied in this paper fundamentally different from the general-purpose coding tasks? To what extent is the medical knowledge essential here? If an agent excels in general-purpose coding tasks, does it still perform well here? How do the rankings differ?
W2: Similarly, the related work lacks a discussion of existing general-purpose coding benchmarks.
W3: The data construction step in Section 3.2 is unclear, and particularly, it does not show the difference between the contributed benchmark and the constituent datasets from existing work. Based on Table 2, is MedAgentGym simply an ensemble of all the existing benchmark datasets? Is this paper overclaimed?
W4: Can you provide some example instances to illustrate MedAgentGym qualitatively?

**Questions:**

Q1: In Line 358, how exactly do you prepare the online pairs for DPO? Isn't DPO an offline algorithm?
Q2: What do you mean by "accurately selects successful trajectories" in Line 417? Can you explain the difference between Pass@K and Best@K? Are they metrics for the agent or for the verifier? And why does a small gap between the two metrics indicate that "the verifier can effectively identify successful trajectories"?
Q3: What do you mean by "repeat this DPO step iteratively" in Line 443, and what do you mean by "DPO using eight new rollouts per task"? Can you explain the setup of iDPO? And why do you need "eight new rollouts per task" in addition to the 4,298 pairs?
Q4: Can you compare the results from the self-improvement in Section 5.3 and the results using the setup in Section 5.1?
Q5: How exactly is Figure 10 computed? Based on what features did you calculate the cosine similarity, and what does the "in/out-of-distribution" in "inter-distribution" mean? How did you determine if a task should belong to in- or out-of-distribution? What's the average number of turns and other statistics of those tasks?

---

> ### Author Response · Authors · 2025-11-21
> **Response to Reviewer Keuw -- Part I**
>
> We are grateful for the reviewer’s constructive remarks and suggestions. We then respond to each point in the following.
>
> ***
> > W1.1: How are the tasks studied in this paper fundamentally different from the general-purpose coding tasks? To what extent is medical knowledge essential here?
>
> **A**: We thank the reviewer for raising this question about the distinct nature of our benchmark. Tasks in MedAgentGym differ fundamentally from generic coding benchmarks in that correctness is governed by biomedical knowledge and clinical standards as well as by programming syntax. **First, domain knowledge is required to specify the target computation:** a common failure mode for strong coding models is to implement outdated or clinically incorrect formulas despite syntactically valid code. For example, Figure 17 shows a base model using the obsolete MDRD equation, whereas Med‑Copilot correctly implements the 2021 CKD‑EPI guideline. **Second, the task distribution is intrinsically biomedical focus:** MedAgentGym spans clinical database querying, clinical note–table consistency checking, medical calculations, FHIR/health‑IT operations, bioinformatics pipelines, biomedical data analysis, biostatistics, and EHR‑based ML, all of which depend on EHR schemas, clinical rules, domain‑specific libraries, and statistical rigor that are not exercised in standard coding benchmarks.
>
> > W1.2: If an agent excels in general-purpose coding tasks, does it still perform well here? How do the rankings differ?
>
> **A**: We thank the reviewer for this important question. Empirically, we find that strong general‑purpose coding baselines do not guarantee strong performance on MedAgentGym.
> In our original leaderboard, we included several coding‑specialized LLMs, such as **codex‑mini**, **Qwen2.5‑Coder‑7B‑Instruct** and **‑14B‑Instruct** (Table 3); We have also added a very recent strong coding model, **Seed‑Coder‑8B‑Reasoning \[1\].**
>
> |  | mimiciii | eicu | treqs | medcalcbench | medagentbench | biocoder | biodsbench | ehrshot | Avg. |
> | :------------------ | :---: | :---: | :---: | :---: | :---: | :---: | :---: | :---: | :---: |
> | gpt-4.1 | 69.36 | 64.75 | 74.97 | 86.23 | 57.63 | 52.95 | 67.35 | 87.93 | 70.15 |
> | Codex-mini | 67.30 | 64.75 | 74.57 | 82.49 | 58.76 | 48.78 | 67.64 | 58.76 | 65.38 |
> | Qwen2.5-7B-Instruct | 13.08 | 15.57 | 12.76 | 25.91 | 30.36 | 21.79 | 10.20 | 5.42 | 17.43 |
> | Qwen2.5-Coder-7B-Instruct | 9.12 | 10.66 | 15.63 | 24.62 | 18.75 | 10.60 | 17.24 | 10.55 | 14.65 |
> | **Seed-Coder-8B-Reasoning \[1\]** | 42.51 | 45.74 | 39.50 | 35.18 | 28.81 | 23.72 | 20.41 | 22.89 | 32.35 |
> | MedCopilot-7B | 68.21 | 68.73 | 70.50 | 92.33 | 55.87 | 37.40 | 71.11 | 33.18 | 62.17 |
> | Qwen2.5-14B-Instruct | 17.21 | 14.07 | 16.43 | 27.40 | 35.59 | 29.49 | 16.33 | 4.45 | 20.12 |
> | Qwen2.5-Coder-14B-Instruct | 41.82 | 44.26 | 35.78 | 33.75 | 30.42 | 26.28 | 22.45 | 28.37 | 32.89 |
> | MedCopilot-14B |  68.78 | 69.34 | 76.84 | 95.81 | 57.41 | 49.32 | 94.78 | 59.05 | 71.42 |
>
> Across these models, we consistently observe that coding‑tuned LLMs provide only limited improvements and often underperform their general‑purpose backbones and our Med‑Copilot agents on MedAgentGym, likely due to the difference (e.g., task format, domain knowledge) between generic coding benchmarks and our biomedical, data‑science–oriented tasks. A similar pattern holds for medical QA‑specialized LLMs (e.g., HuatuoGPT‑o1‑7B, MedReason‑8B, Baichuan‑M1‑14B). Taken together, these results indicate that MedAgentGym emphasizes a joint capability, robust code generation plus biomedical semantics and data‑science practice, rather than pure coding or pure medical QA in isolation.
>
> [1] Seed, ByteDance, et al. "Seed-coder: Let the code model curate data for itself." arXiv preprint arXiv:2506.03524 (2025).

---

> > ### Author Response · Authors · 2025-11-21
> > **Response to Reviewer Keuw -- Part II**
> >
> > > W2: Similarly, the related work lacks a discussion of existing general-purpose coding benchmarks.
> >
> > A: Thank you for raising this important point. While several benchmarks exist for general-purpose coding benchmarks (e.g., SWE-bench, AgentBench), MedAgentGYM is fundamentally distinct in both domain focus and design:
> >
> > - **Domain-specific coding focus**: Unlike general coding agent benchmarks that target software engineering tasks, MedAgentGYM emphasizes biomedical coding reasoning, requiring integration of clinical knowledge and domain-specific data formats (e.g., EHRs, lab reports, biological sequences) within executable environments.
> >
> > - **Executable and interactive environment**: In contrast to existing medical benchmarks which are typically static and QA-based, MedAgentGYM provides isolated Docker containers, interactive feedback, and support for trajectory sampling and debugging. This enables training and evaluation of code-generating medical agents in a realistic setting.
> >
> > - **Training-oriented infrastructure**: While prior work focuses primarily on benchmarking (e.g., MedAgentBench, MedCalcBench), MedAgentGYM uniquely supports systematic training of LLM agents, with over 6,000 released trajectories, interactive debugging, and a modular scaffold for reinforcement learning.
> >
> > - **Scalability and diversity**: MedAgentGYM includes over 72,000 tasks across 129 categories derived from 12 real-world biomedical scenarios, making it significantly broader in scope compared to prior medical coding datasets.
> >
> > > W3: The data construction step in Section 3.2 is unclear, and particularly, it does not show the difference between the contributed benchmark and the constituent datasets from existing work. Based on Table 2, is MedAgentGym simply an ensemble of all the existing benchmark datasets? Is this paper overclaimed?
> >
> > **A**: Thank you for the insightful questions. MedAgentGym is not a simple ensemble of existing datasets; rather, we transform and unify heterogeneous sources into a single executable training environment with (i) Docker‑based isolation, (ii) a common JSON I/O schema, (iii) natural‑language error grounding, and (iv) execution‑verified outcome checks (Section 3.2, Appendix C.5). Specifically, MedAgentGym features:
> >
> > * **From static benchmarks to verifiable coding tasks:** Existing benchmarks typically provide static QA pairs or code snippets. We re-engineer them into dynamic tasks with **verifiable execution-based ground truths**. (1) *From Narrative to Executable (e.g., MedCalcBench):* We transform static medical calculation problems into programming tasks, requiring agents to implement formulas via code rather than merely retrieving values or text. (2) *From Reference Code to Execution Verification (e.g., BioCoder):* Where only reference implementations are available, we build execution pipelines to derive definitive output signatures, enabling robust functional verification instead of fragile text matching. (3) *Procedural Expansion (e.g., N-PowerAI):* For small-scale tasks like biostatistics, we apply attribute substitution to expand each task type 100×, recomputing targets to create sufficient volume for training and evaluation. (4) *Structured Re-linking (e.g., EHRCon):* For tasks including unstructured metadata, we rebuild disparate metadata and clinical notes into a unified, linked database structure with explicit inconsistency labels to support complex querying.
> > * **Interactive, privacy‑preserving execution environment:** Beyond data curation, MedAgentGym introduces a Docker‑based sandbox for each task, with pre‑installed biomedical libraries and a terminal tool for dynamic dependency installation. All code runs in this isolated environment, with compile/runtime errors normalized into unified natural‑language messages that LLMs can use for interactive debugging. This design is absent from the source datasets, which are original static and non‑executable.
> > * **Trajectory and verifier infrastructure for agent training:** Building on this environment, we implement multi‑threaded rollout backends (Ray/Joblib) and release \~6K successful and preference trajectories plus an outcome‑supervised verifier used both for evaluation and as a reward model in RL. None of these original datasets provides agent‑level interaction logs, trajectory structures, or reward models; these are introduced by MedAgentGym specifically to make the benchmark a training ground for agentic RL, not merely an evaluation suite.
> >
> > We will emphasize these distinctions in Section 3.2 and Appendix C.5 by explicitly summarizing, for each source dataset, the task transformations and environment‑level additions introduced by MedAgentGym.

---

> > > ### Author Response · Authors · 2025-11-21
> > > **Response to Reviewer Keuw -- Part III**
> > >
> > > > W4: Can you provide some example instances to illustrate MedAgentGym qualitatively?
> > >
> > > **A**: We appreciate your suggestion regarding concrete examples to ground the quantitative results. We highlight four representative case studies in **Appendix F.6 (Figures. 14–17)** spanning bioinformatics, numerical optimization, and clinical calculation, which illustrate both characteristic failure modes and how MedAgentGym’s interactive environment enables correction:
> > >
> > > * **Interactive Debugging (Figure 14):** We demonstrate a bioinformatics task (BioCoder) where the agent initially fails with an AttributeError. Unlike static models, the MedAgentGym environment provides runtime traceback feedback, allowing the agent to actively debug and correct the API usage to reach a successful solution.
> > > * **Domain-Specific Correctness (Figures 15 & 17):** We highlight critical "silent failures" where baselines generate syntactically valid but medically incorrect code. For instance, Figure 17 shows a baseline model defaulting to an outdated GFR formula (MDRD), whereas Med-Copilot correctly implements the complex, conditional logic of the modern 2021 CKD-EPI standard.
> > > * **Optimization Semantics (Figure 16):** We contrast a baseline attempting to define an optimization objective using a simple Python list (mathematically invalid) against Med-Copilot’s correct usage of linear expression objects, demonstrating superior grasp of library-specific semantics.
> > >
> > > These qualitative cases complement the aggregate metrics by showing that (i) interactive error grounding materially improves iterative code refinement, and (ii) biomedical tasks demand domain-aware implementations and up‑to‑date clinical formulas rather than merely syntactic correctness. We will include more qualitative examples to broaden coverage across task families in the revision.
> > >
> > > > Q1: In Line 358, how exactly do you prepare the online pairs for DPO? Isn't DPO an offline algorithm?
> > >
> > > **A**: We apologize for the typo here. The two types of data we use are actually: (1) off-policy preference pairs obtained from GPT-4.1-mini, and (2) on-policy preference pairs obtained from the current policy model. We will revise the manuscript to correct with the standard terms “off-policy” and “on-policy”.
> > >
> > > > Q2.1: What do you mean by "accurately selects successful trajectories" in Line 417?
> > >
> > > **A**: Thank you for your question. By “accurately selects successful trajectories,” we refer to the ability of our trained verifier to pick a trajectory that actually solves the task from a set of candidate generations. Concretely, given a set of trajectories produced by the agent, the verifier assigns each trajectory a scalar score that estimates its likelihood of success. We rank all trajectories by this score and select the top-ranked one as the predicted successful trajectory. We then execute this selected trajectory and check whether it indeed succeeds. The reported ratio is the fraction of instances for which the verifier’s top-ranked trajectory is truly successful, i.e., the verifier’s accuracy in selecting successful trajectories. We will revise the original text to improve clarity.
> > >
> > > > Q2.2: Can you explain the difference between Pass@K and Best@K? Are they metrics for the agent or for the verifier?
> > >
> > > **A**: Pass@K and Best@K capture different components of the agent–verifier pipeline. Pass@K is an agent-only metric: it measures the probability that at least one of the K trajectories sampled from the agent is successful, without involving the verifier. In contrast, Best@K evaluates the verifier’s selection behavior: given the same K trajectories, the verifier ranks them using its predicted success scores, and we check whether the top-ranked trajectory is actually successful. Thus, Pass@K reflects the agent’s ability to produce successful trajectories, while Best@K reflects the verifier’s ability to identify a successful trajectory when one exists.
> > >
> > > > Q2.3: Why does a small gap between the two metrics indicate that "the verifier can effectively identify successful trajectories"?
> > >
> > > **A**: By definition, Best@K cannot exceed Pass@K, since Best@K requires (i) that the agent has produced at least one successful trajectory (counted in Pass@K) and (ii) that the verifier assigns it the highest score. A small gap between Pass@K and Best@K therefore indicates that, whenever the agent produces a successful trajectory, the verifier almost always ranks it first. Conversely, a large gap would indicate that the agent often produces correct solutions that the verifier fails to select. Hence, a small Pass@K–Best@K gap empirically supports our claim that the verifier can effectively identify successful trajectories.

---

> > > > ### Author Response · Authors · 2025-11-21
> > > > **Response to Reviewer Keuw -- Part IV**
> > > >
> > > > > Q3: What do you mean by "repeat this DPO step iteratively" in Line 443? What do you mean by "DPO using eight new rollouts per task"? Can you explain the setup of iDPO? And why do you need "eight new rollouts per task" in addition to the 4,298 pairs?
> > > >
> > > > **A**: Thank you for this important question. iDPO differs from standard DPO in that DPO performs a single offline preference-based update on a fixed dataset, whereas iDPO iteratively alternates between collecting new on‑policy preference pairs and running DPO updates so that the policy and training data co‑evolve. Beyond expert‑generated trajectories from gpt-4.1-mini (Section 5.1), we also explore self‑improvement by refining the model on its own outputs (Qwen2.5-7B-Instruct, Section 5.3). We first apply rejection‑sampling SFT: starting from Qwen2.5-7B-Instruct, we collect 1,000 successful trajectories and perform filtered behavior cloning on this set. We subsequently apply DPO using on-policy preference pairs generated by the rejection sampling SFT checkpoint. Specifically, we sample eight rollouts per task, scoring them via the verifier in Section 5.2, and form 4,298 pairs by contrasting the highest-scoring correct and lowest-scoring incorrect trajectories. Following [1], we repeat this data collection and policy update cycle (iDPO) for further refinement, resampling trajectories and reconstructing preference pairs after each DPO update. We will revise the iDPO setup in the revised manuscript with methodology reference [1] to improve clarity.
> > > >
> > > > [1] Pang, Richard Yuanzhe, et al. "Iterative reasoning preference optimization." Advances in Neural Information Processing Systems 37 (2024): 116617-116637.
> > > >
> > > > > Q4: Can you compare the results from the self-improvement in Section 5.3 and the results using the setup in Section 5.1?
> > > >
> > > > **A**: Thank you for this insightful question. A direct comparison between the expert-supervised (Section 5.1) and self-improvement (Section 5.3) paradigms reveals a significant performance disparity under the SFT → DPO framework. As detailed in the table below, the self-improvement setup is markedly weaker than the setup supervised by gpt-4.1-mini:
> > > >
> > > > |  | mimic-iii | eicu | treqs | medcalcbench | medagentbench | biocoder | biodsbench | ehrshot | Avg |
> > > > | :---- | :---: | :---: | :---: | :---: | :---: | :---: | :---: | :---: | :---: |
> > > > | \+rejection sampling | 35.63 | 37.38 | 33.47 | 51.34 | 41.62 | 28.75 | 34.17 | 8.62 | 33.87 |
> > > > | \+SFT  | 57.83 | 61.48 | 72.66 | 89.06 | 50.85 | 28.33 | 55.10 | 15.62 | 53.87 |
> > > > | **\+ rejection sampling \+ DPO (section 5.3)** | 36.63 | 38.42 | 34.55 | 41.67 | 39.21 | 31.42 | 50.19 | 10.17 | 35.28 |
> > > > | **\+ SFT \+ DPO (section 5.1)** | 68.21 | 68.73 | 70.5 | 92.33 | 55.87 | 37.4 | 71.11 | 33.18 | 62.17 |
> > > >
> > > > Under the expert-supervised regime (Section 5.1), behavioral cloning from gpt-4.1-mini establishes a robust SFT baseline (53.87), which DPO further elevates to 62.17. Conversely, in the self-improvement regime (Section 5.3), the base model (Qwen-2.5-7B-Instruct) exhibits weaker performance on specialized medical coding tasks. Consequently, the trajectories generated for rejection sampling and preference construction are of lower quality. In this setting, the rejection-sampling baseline achieves only 33.87 average, and the addition of DPO yields a marginal gain of 1.4 points (to 35.28).
> > > >
> > > > This discrepancy reflects the supervision quality: gpt‑4.1‑mini supplies high‑quality, domain‑appropriate trajectories that produce a strong SFT initialization for DPO, whereas self‑generated trajectories from the relatively weak base model are much noisier, leading to noisier SFT data and preference pairs. Consequently, these results empirically support our claim that iDPO‑based self‑improvement is strongly bottlenecked by the strength of the initial expert model and the resulting SFT data.

---

> > > > > ### Author Response · Authors · 2025-11-21
> > > > > **Response to Reviewer Keuw -- Part V**
> > > > >
> > > > > > Q5.1: How exactly is Figure 10 computed? Based on what features did you calculate the cosine similarity, and what does the "in/out-of-distribution" in "inter-distribution" mean?
> > > > >
> > > > > **A**: Figure 10 is obtained via a similarity analysis over trajectory embeddings using cosine similarity. Concretely, we proceed in two steps: (1) Trajectory representation and (2) Cosine similarity and distribution plots.
> > > > >
> > > > > First, we use the off‑the‑shelf embedding model DRAGON \[1\] to encode each sampled code trajectory into a fixed‑dimensional vector. Thus, every trajectory (i.e., a full generated code solution for a given task) is mapped to a single dense embedding. We then compute pairwise cosine similarities between these trajectory embeddings under two settings:
> > > > >
> > > > > * **Intra‑category analysis (structured vs. open‑ended):** We partition tasks into two categories: structured and open‑ended. For the red distribution, we randomly sample trajectory pairs within the structured category and compute cosine similarity between their embeddings. For the purple distribution, we randomly sample trajectory pairs across categories (one structured, one open‑ended) and again compute cosine similarity. The resulting density plots exhibit distinct concentration patterns, indicating that trajectories from different task categories occupy different regions in the embedding space.
> > > > > * **Inter‑distribution analysis (in‑distribution vs. out‑of‑distribution):** Following Section 5.4, we treat tasks appearing in the training set as in‑distribution and tasks from external evaluation benchmarks as out‑of‑distribution. Using this partition, we compute cosine similarities in two ways: (i) between trajectory embeddings drawn from the same distribution (in–in or out–out), and (ii) between trajectory embeddings drawn across distributions (in–out). In Figure 10, the red density corresponds to similarities within the same distribution (intra‑distribution), while the purple density corresponds to similarities across in‑distribution and out‑of‑distribution tasks (inter‑distribution). The clear separation between these densities shows that in‑distribution and out‑of‑distribution tasks form noticeably different clusters in the embedding space.
> > > > >
> > > > > \[1\] Lin, Sheng‑Chieh, et al. “How to train your dragon: Diverse augmentation towards generalizable dense retrieval.” arXiv preprint arXiv:2302.07452 (2023).
> > > > >
> > > > > > Q5.2: How did you determine if a task should belong to in- or out-of-distribution? What's the average number of turns and other statistics of those tasks?
> > > > >
> > > > > **A**: We define as *in‑distribution* all tasks whose instances are used for SFT or RL training, and as *out‑of‑distribution* all tasks drawn from external evaluation benchmarks that are never seen during training; the full dataset inventory and sizes for both groups are listed in Table 2\. To further characterize this split for in- and out-of-distribution tasks, we compute summary statistics over gpt‑4.1 trajectories (average turns and code‑level metrics: complexity, maintainability, lines of code, and logical lines of code):
> > > > >
> > > > > | gpt-4.1 | mimiciii | eicu | treqs | medcalcbench | medagentbench | biocoder | biodsbench | ehrshot | IDavg | EHRSeqSQL | EHRcon | MIMICExtract | NpowerAI | OODavg |
> > > > > | :---- | :---: | :---: | :---: | :---: | :---: | :---: | :---: | :---: | ----- | ----- | :---: | :---: | :---: | ----- |
> > > > > | **\# turns** | 25.91 | 26.59 | 20.65 | 10.73 | 17.28 | 22.08 | 21.75 | 8.71 | 19.21 | 25.83 | 38.97 | 10.42 | 22.64 | 20.83 |
> > > > > | **complexity** | 0.01 | 0.06 | 0.01 | 4.09 | 0.23 | 7.77 | 0.17 | 20.97 | 4.16 | 0.03 | 0.11 | 20.76 | 0.04 | 4.49 |
> > > > > | **maintainability** | 95.14 | 95.99 | 96.62 | 88.38 | 91.04 | 68.20 | 92.67 | 56.24 | 85.54 | 94.17 | 92.65 | 64.82 | 96.94 | 86.03 |
> > > > > | **loc** | 9.26 | 9.67 | 4.17 | 19.00 | 18.89 | 24.82 | 28.97 | 144.69 | 32.43 | 9.44 | 12.45 | 129.76 | 5.24 | 34.52 |
> > > > > | **lloc** | 5.86 | 6.33 | 3.00 | 15.20 | 10.79 | 21.84 | 16.44 | 110.51 | 23.75 | 6.21 | 8.90 | 117.54 | 3.73 | 26.93 |
> > > > >
> > > > > From the comparison, we observe that in‑ and out‑of‑distribution averages are relatively similar. This suggests that the observed distribution shift cannot be explained by trivial surface properties such as code length or basic complexity, but instead reflects deeper differences in task type and intrinsic difficulty. We will update Table 8 in the updated manuscript to include additional evaluation of code quality and efficiency for out-of-distribution tasks.
> > > > >
> > > > > ***
> > > > > Thank you once again for your insightful review. We appreciate your feedback on our work. Feel free to let us know if you have any further questions, and we are happy to discuss further.

---

> > > > > > ### Comment · Reviewer_Keuw · 2025-11-26
> > > > > >
> > > > > > Thanks for the detailed reply. Most of my concerns are addressed. I'll update the score accordingly.

---

> > > > > > > ### Author Response · Authors · 2025-11-26
> > > > > > >
> > > > > > > Thank you so much for reading our rebuttal! We are glad that our responses have addressed your concerns. We will make sure to include those points in our next version.

---

### Official Review · Reviewer_xZ94 · 2025-10-23

**Soundness:** 2
**Presentation:** 2
**Contribution:** 2
**Rating:** 4
**Confidence:** 3

**Summary:**

This paper presents MedAgentGym, an extensible agentic environment and benchmark designed to systematically advance code-centric medical reasoning in LLM-based agents. It encapsulates 72,413 executable medical data science tasks spanning 129 categories derived from 12 scenarios, featuring isolated Docker-based sandboxes, ground-truth verifiers, multi-turn feedback, and scalable trajectory collection. Through comprehensive empirical evaluation, MedAgentGym is used to benchmark 29 LLMs, highlighting persistent deficits for medical code generation, and demonstrating substantial improvements via agentic reinforcement learning fine-tuning.

**Strengths:**

1. MedAgentGym aggregates an exceptionally broad and diverse set of medical code-centric tasks.
2. The environment is built around reproducible, interactive Docker sandboxes, allowing code execution, error handling, debugging, and dynamic dependency install, addressing reproducibility and privacy.
3. Med-Copilot exhibit strong improvements over baselines, with RL strategies yielding notable boosts, and ablation studies clarifying contributions.

**Weaknesses:**

1. MedAgentGym is constructed by integrating 12 existing datasets. Although it provides a division between training and test sets, the model is exposed to the task types and data patterns from these datasets during training. Therefore, its strong performance on the internal test set may partially result from memorization of specific task patterns or overfitting, rather than genuinely acquiring a universal biomedical code reasoning capability.
2. While integrating these components into a large-scale, biomedical-oriented environment represents a significant engineering contribution, the core technical concepts underlying the environment—such as the use of Docker sandboxes, provision of interactive debugging feedback, and trajectory collection for reinforcement learning—have precedents in the AI agent domain.
3. The medical-specific models evaluated in the paper, such as HuatuoGPT-o1-7B and MedReason-8B, are relatively small, with only 7B/8B parameters. Attributing their suboptimal performance solely to the limitations of medical specialization, while overlooking the substantial differences in model size, is logically flawed. A more equitable comparison would be to assess a large-scale, medically optimized model against a general-purpose large model.

**Questions:**

1. Can the authors clarify how different error types (see Figure 7) are incorporated into the RL reward signal? Are there distinct penalties for, say, 'stuck in the loop' vs. compile/runtime/IO errors? How sensitive is final agent performance to this reward model?

---

> ### Author Response · Authors · 2025-11-21
> **Response to Reviewer xZ94 -- Part I**
>
> We are grateful for the reviewer’s constructive remarks and suggestions. We respond to each point in the following.
>
> ***
>
> > W1: Overfitting to Internal Test Set.
>
> **A**: We appreciate the reviewer’s insight regarding the potential for overfitting to specific task patterns. To rigorously evaluate this, **we intentionally designed MedAgentGym with a strict two-tier evaluation protocol that separates Internal (8 of 12 datasets) from External (4 of 12 datasets) validation**.
>
> * **Strict Separation of Unseen Tasks (Table 2\)**: We explicitly held out 4 of the 12 datasets (EHR-SeqSQL, EHRCon, MIMIC-Extract, and N-PowerAI) to serve as a dedicated External Validation set. The inter‑distribution trajectory analysis shows substantial divergence between internal and external tasks (Figure 10 (b)). These external datasets were completely excluded from the training pipeline and distinct to training tasks (e.g., sequential SQL, hybrid text–note inconsistency detection, raw data, biostatistical test), ensuring that the model is never exposed to their specific data or task patterns during post-training.
> * **Empirical Evidence of Transferable Reasoning (Table 5\)**: Our results on these unseen external datasets demonstrate robust generalization rather than simple memorization. As shown in Table 5, Med-Copilot-14B (GRPO) achieves an average score of 47.02% on these held-out tasks, representing a substantial \+19.1% improvement over the base model (Qwen-2.5-14B-Instruct). This significant gain on unseen distributions confirms that MedAgentGym instills universal, transferable biomedical coding capabilities that extend beyond the training tasks.
>
> We will add additional analysis of external validation in the updated manuscript.
>
> > W2: Novelty of MedAgentGym.
>
> **A**: Thank you for raising this important point. While several benchmarks exist for general-purpose agent evaluation (e.g., SWE-bench, AgentBench) or natural language-based medical reasoning (e.g., MedQA, PubMedQA), MedAgentGYM is unique in both domain focus and design:
>
> - **Domain-specific coding focus**: Unlike general coding agent benchmarks that target software engineering tasks, MedAgentGYM emphasizes biomedical coding reasoning, requiring integration of clinical knowledge and domain-specific data formats (e.g., EHRs, lab reports, biological sequences) within executable environments.
>
> - **Executable and interactive environment**: In contrast to existing medical benchmarks which are typically static and focus on text-based QA tasks only, MedAgentGYM provides isolated Docker containers, interactive feedback, and support for trajectory sampling and debugging. This enables training and evaluation of code-generating medical agents in a realistic setting.
>
> - **Training-oriented infrastructure**: While prior work focuses primarily on benchmarking (e.g., MedAgentBench, MedCalcBench), MedAgentGYM uniquely supports systematic training of LLM agents, with over 6,000 released trajectories, interactive debugging, and a modular scaffold for reinforcement learning.
>
> - **Scalability and diversity**: MedAgentGYM includes over 72,000 tasks across 129 categories derived from 12 real-world biomedical scenarios, making it significantly broader in scope compared to prior medical coding datasets.
>
> > W3: Large-scale Medical Domain-specific Models.
>
> **A**: Thank you for the question. Following your suggestion, we have added an additional experiment using large-scale medical domain-specific reasoning models, including **HuatuoGPT-o1-72B \[1\] and Baichuan-M2-32B \[2\]:**
>
> |  | mimiciii | eicu | treqs | medcalcbench | medagentbench | biocoder | biodsbench | ehrshot | Avg. |
> | :---------- | :-------: | :---: | :---: | :---: | :---: | :---: | :---: | :---: | :---: |
> | Qwen2.5-7B-Instruct | 13.08 | 15.57 | 12.76 | 25.91 | 30.36 | 21.79 | 10.20 | 5.42 | 17.43 |
> | HuatuoGPT-o1-7B  | 4.99 | 7.04 | 7.04 | 38.05 | 18.64 | 28.21 | 19.88 | 5.03 | 16.11 |
> | Qwen2.5-14B-Instruct | 17.21 | 14.07 | 16.43 | 27.40 | 35.59 | 29.49 | 16.33 | 4.45 | 20.12 |
> | Baichuan-M1-14B | 4.50 | 12.19 | 7.36 | 1.82 | 21.46 | 16.34 | 17.42 | 0.00 | 10.14 |
> | Qwen2.5-32B-Instruct | 54.56 | 45.41 | 62.81 | 69.96 | 40.67 | 27.45 | 22.45 | 18.13 | 42.68 |
> | **Baichuan-M2-32B \[2\]** | 20.83 | 23.61 | 24.92 | 30.02 | 25.42 | 25.00 | 20.41 | 12.94 | 22.89 |
> | Llama-3.3-70B-Instruct | 39.93 | 25.08 | 24.98 | 84.99 | 39.40 | 27.55 | 24.49 | 29.93 | 37.04 |
> | DeepSeek-R1-Distill-Llama-70B | 64.59 | 64.92 | 56.98 | 76.96 | 28.81 | 32.05 | 42.86 | 33.42 | 50.07 |
> | **HuatuoGPT-o1-72B \[1\]** | 27.19 | 29.84 | 29.65 | 52.01 | 28.81 | 31.41 | 26.53 | 16.87 | 30.29 |

---

> > ### Author Response · Authors · 2025-11-21
> > **Response to Reviewer xZ94 -- Part II**
> >
> > **(cont'd)** We observe that large-scale medical reasoning models exhibit similar patterns to its 7B/8B counterparts. While its absolute performance is higher than the smaller models, it still significantly underperforms generalist models on coding-centric biomedical tasks. Medical specialization (even at 72B parameters) improves narrative reasoning but does not inherently translate to the computational and coding proficiency required for tasks in MedAgentGym. We will include the new experimental results of large-scale medical reasoning models in the final manuscript to provide a more equitable and rigorous comparison.
> >
> > [1] Chen, Junying, et al. "Huatuogpt-o1, towards medical complex reasoning with llms." arXiv preprint arXiv:2412.18925 (2024).
> > [2] Dou, Chengfeng, et al. "Baichuan-m2: Scaling medical capability with large verifier system." arXiv preprint arXiv:2509.02208 (2025).
> >
> > > Q1: Error Analysis and Reward Design.
> >
> > **A**: We thank the reviewer for this insightful query regarding reward composition. We wish to clarify that the error distribution in Figure 7 represents a post-hoc analysis of the strongest baseline (gpt-4.1) to identify future improvement directions, rather than a blueprint for our reward design. **Our primary RL training utilizes a sparse, outcome-centric reward (success=1, failure=0) without manual engineering for specific error types.**
> >
> > Motivated by the reviewer’s question, we introduced a penalty term (r\_penalty \=−0.3) triggered when the agent generated repetitive code blocks (defined as cosine similarity \>0.9 between consecutive code generations). As shown in the table below, introducing this heuristic penalty resulted in a consistent performance degradation across all datasets, dropping the average score from 62.17% to 56.98%. Manual inspection of trajectories suggests that the penalty suppresses benign self‑debugging and iterative refinement (e.g., small edits to a previous code block), leading the agent to terminate early or to change strategy prematurely rather than repairing its own code.
> >
> > |  | mimiciii | eicu | treqs | medcalcbench | medagentbench | biocoder | biodsbench | ehrshot | Avg. |
> > | :------------- | :---: | :---: | :---: | :---: | :---: | :---: | :---: | :---: | :---: |
> > | GRPO | 68.21 | 68.73 | 70.50 | 92.33 | 55.87 | 37.40 | 71.11 | 33.18 | 62.17 |
> > | **GRPO w/ penalty** | 63.68 | 65.08 | 64.82 | 84.13 | 52.54 | 33.33 | 65.31 | 26.97 | 56.98 |
> >
> > We will add the experimental results and the discussion of optional reward design (e.g., small penalties for repeated identical actions) in Appendix F.9 to guide users who prefer more granular rewards; these are implementation‑level options, not prerequisites for the reported gains.
> >
> > ***
> > Thank you once again for your insightful review. We appreciate your feedback on our work. Feel free to let us know if you have any further questions, and we are happy to discuss further.

---

> ### Comment · Reviewer_xZ94 · 2025-11-26
>
> Thank you for the detailed rebuttal. The new experiments have addressed my main concerns. I have raised my score accordingly.

---

> > ### Author Response · Authors · 2025-11-26
> >
> > Thank you for your insightful suggestions and questions! We are glad that our responses have addressed your concerns. We will include all newly added experiments and discussions in our next version.

---

### Official Review · Reviewer_cegQ · 2025-10-28

**Soundness:** 3
**Presentation:** 3
**Contribution:** 3
**Rating:** 8
**Confidence:** 2

**Summary:**

The paper presents a released training environment MedAgentGym for the purposes of benchmarking and training LLMs for the use of biomedical data science coding tasks. Benchmarking against many propriatary and (varying sized) open-source LLMs demonstrates the state of the field in this task. Training demonstrates impressive gains in task performance of OS LLMs, comparable to gpt-4o. The authors conduct extensive benchmarking and experiments to demonstrate the utility of their training environment.

**Strengths:**

* The paper is written to a good standard and certainly looks publication-ready
* The consideration of open-source LLMs for the papers setting of biomedical data science is important, as a large amount of data will be under stringent data privacy rules. Many alternative similar papers operating in this field do not consider this
* Good contribution over prior literature, encapsulating the majority of tasks that I believe would be applicable for biomedical data science
* Extensive benchmarking of many existing open-source and proprietary LLMs gives important information regarding the capabilities of such models
* Thorough experiments for model Med-Copilot. Results look promising

**Weaknesses:**

* I have a feeling that the the title and naming given to the training environment is slightly overstepping and too generelized. Perhaps 'BioMedAgentGym' is more suitable.
* Since the paper is releasing a training environment for the practical real-world use of biomedical data science, I would like to see some discussion on the implications of this and reccomendations to users (please see questions below)
* I cannot see many weakensses, though I am not familiar with the field of biomedical research nor such benchmarking papers

**Questions:**

* Table 3 demonstrates results of LLMs on MedAgentGym. Some "best avg. scores" (for a given LLM size) are relatively quite low. For example, the OSS <10B has Qwen3-8B at a success rate of 30.83. Given that some practitioners may only have the compute for such models, are you able to give reccomendations (complementing the writing in §4.2) regarding this? For example, what do you deem a sufficiently good performance on benchmarking for real-world deployment of a LLM?

---

> ### Author Response · Authors · 2025-11-21
> **Response to Reviewer cegQ**
>
> Thank you so much for your positive and encouraging feedback. Below, we address your comments and suggestions in details:
>
> ***
>
> > W1: Better Name as BioMedAgentGym.
>
> **A**: We thank the reviewer for this thoughtful suggestion. We agree that 'BioMedAgentGym' more precisely reflects the benchmark's extensive scope, which bridges both clinical medicine and broader biomedical data science (e.g., bioinformatics and biostatistics). For the duration of this rebuttal, we have retained the original naming convention to ensure consistency with the submitted manuscript and associated discussions. **We will update the terminology in the final camera-ready version to reflect this change**.
>
> > W2 & Q1.1: Implications for Low-Resource Settings.
>
> **A**: We thank the reviewer for highlighting the practical constraints of deploying biomedical agents in resource-limited environments. Our core recommendations for leveraging MedAgentGym in low-resource settings are as follows:
>
> * **Prioritizing Agentic Fine-Tuning over Zero-Shot Inference.** While \<10B OSS models exhibit low zero-shot performance (e.g., Qwen3-8B at \~30%), ***MedAgentGym is designed specifically to bridge this gap through agentic training*****.** As shown in Table 4, agentic fine-tuning (e.g., GRPO) boosts the performance of a lightweight 7B model from 16.89% to 62.17%, effectively matching the performance of commercial APIs (e.g., gpt-4.1-mini at 58.76%). For practitioners, this confirms that small models are viable for deployment if fine-tuned rather than prompted zero-shot.
> * **Reducing Computational Barriers via Released Training Resources.** To support practitioners who lack the computational resources for extensive exploration, we have publicly released: (1) **Med-Copilot**: A ready-to-use, fine-tuned 7B/14B model that achieves state-of-the-art performance among open-source models. (2) **6K+ High-Quality Trajectories**: We release the successful training trajectories (Section 5.1). This allows users to perform SFT & DPO immediately without incurring the high computational cost of sampling and filtering trajectories from scratch.
> * **Inference-Time Optimization with Verifiers**. For deployment, we recommend integrating the Outcome Verifier we developed (Section 5.2). Our analysis shows that a small model combined with a verifier (Best@K) significantly improves reliability. This allows low-resource users to trade a small amount of inference-time compute (sampling a few solutions) for a large gain in accuracy, avoiding the need for massive proprietary models.
> * **Task-Specific Specialization**. The "average scores" in Table 3 reflect a generalist capability across 129 diverse categories in 8 datasets. However, fine-tuning on single, specific downstream tasks exclusively may enable higher performance than the aggregate benchmark suggests. Practitioners should fine-tune small models specifically on the subset of MedAgentGym relevant to their target application to maximize efficiency.
>
> We will add the recommendation for practitioners regarding the low-resource settings in the updated manuscript.
>
> > Q1.2: Defining "Sufficient Performance" for Real-World Deployment.
>
> **A**: Thank you for this important question. Defining a deployment threshold is context-dependent, but we advocate for benchmarking against human expert performance and commercial SOTA. In our human study (Table 12), **domain experts achieved an average success rate of 75% on structured tasks and 45% on open-ended tasks**. Therefore, we consider a model score in the 60-65% range (achieved by Med-Copilot and gpt-4o) as a "sufficient" threshold for deployment as *a human-in-the-loop assistant*. Given that even strong models (and humans) do not achieve 100% accuracy, we recommend deploying these models strictly as "Copilots" rather than fully autonomous agents. The environment's interactive features (e.g., debugging, verifiable execution) are designed to facilitate this oversight, allowing human experts to validate code execution outputs rather than manually writing code from scratch. We will add the discussion regarding the real-world deployment in the updated manuscript.
>
> ***
> Thank you again for your review. We hope our response could address your concerns. If you have any further questions, we would be happy to discuss them further.

---

> ### Comment · Reviewer_cegQ · 2025-11-24
>
> Thanks for your response. I'm satisfied with the response to my questions. I think these points are important for users connecting the paper to practical reality.
>
> Additionally, after having read the other reviews and corresponding rebuttals I am increasingly confident that this paper should be accepted. I will upgrade my confidence score to reflect this.
>
> As a friendly point, I have recently came across this NeurIPS 25 publication of which serves as a "benchmark of benchmarks": https://arxiv.org/pdf/2511.04703 . Since this is a very recent publication, I do not think it is fair to expect the authors to have included the findings of this paper in analyzing the efficacy of their work. However, in the spirit of "good science", it could be good to do so if the paper is accepted.

---

> > ### Author Response · Authors · 2025-11-24
> >
> > We truly appreciate your constructive suggestions and thoughtful discussion, which greatly helped us improve the quality of our manuscript. We are very glad to hear that our revisions have satisfactorily addressed your concerns. Following your suggestion , we will explicitly discuss MedAgentGym through the lens of the construct-validity framework (phenomenon–task–metric–claims) [1] in the updated manuscript. Thank you again for your insightful feedback and comments!
> >
> > [1] Bean, Andrew M., et al. "Measuring what Matters: Construct Validity in Large Language Model Benchmarks." NeurIPS (2025).

---

### Official Review · Reviewer_7VJa · 2025-10-30

**Soundness:** 4
**Presentation:** 4
**Contribution:** 4
**Rating:** 8
**Confidence:** 4

**Summary:**

MedAgentGym introduces an interactive training environment for code-centric biomedical reasoning. There are numerous scenarios, with different LLMs evaluated in the environment showing gaps between especially closed vs open models. They also introduce Med-copilot trained on the env which is very strong.

**Strengths:**

1. Solving an important problem & provides a comprehensive benchmark on real-world medical tasks.

2. Very rigorous evals across many tasks over many models

3. Strong Performance Gains of Med-Copilot on the env

**Weaknesses:**

- Only execution evals, no assessment of intermediate reasoning and steps which is vital in medicine. Where the trajectory matters as much as the solution

- Big OOD drops unexplained on external dataset (more validation and digging). Maybe things are just overfit?

**Questions:**

- Can you assess some trajectories for sound reasoning vs just only correctness

- Can we know that none of the LLMs have already trained on the benchmark datasets? Maybe need some private data or new data

- To the OOD drop please can you dig in and understand why?

- Please can you add variance not just mean to understand overlap of models

---

> ### Author Response · Authors · 2025-11-21
> **Response to Reviewer 7VJa -- Part I**
>
> Thank you for your insightful comments and for taking the time to review our paper. We respond to your suggestions point by point as follows.
>
> ***
> > W1 & Q1: Evaluation of intermediate reasoning trajectories.
>
> **A**: Thank you for this valuable suggestion. In Appendix F8, we have performed **a quantitative analysis on 250+ trajectories** (randomly sampled over 10% of our trajectory collection) and confirmed that the vast majority of successful solutions followed a logically sound path, **with cases of 'correct answer from flawed code' being exceptionally rare (\<1%)**. Unlike narrative reasoning tasks (e.g., multiple-choice QA) where flawed logic might coincidentally lead to a correct answer, code-based tasks in MedAgentGym have a strong intrinsic check: **the code must execute correctly to produce a solution**. This makes 'wrong trajectory with correct solution' less likely.
>
> Besides correctness upon execution evaluations, we have further reported additional quality evaluation of intermediate trajectories in Appendix F1, including (1) **number of turns** for interaction effectiveness, (2) **cyclomatic complexity** for code complexity,  (3) **maintainability index** for code readability, and (4) **line-of-code (loc)** and (5) **logical line-of-code (lloc)** for code efficiency. Comparing different models (averaged across datasets), we observe that advanced closed-source models generate more complex and longer code; after training, MedCopilot-7B produces structurally efficient and more maintainable code compared to backbone models.
>
> | Model | # Turns | Complexity | Maintainability | LOC | LLOC |
> |:--------|:----------:|:---------------:|:-------------------:|:------:|:--------:|
> | GPT-4.1 | 20.96 | 4.52 | 86.07 | 34.70 | 27.20 |
> | GPT-4.1-mini | 19.43 | 4.25 | 86.10 | 40.32 | 28.63 |
> | Qwen-2.5-7B-Instruct | 15.98 | 2.71 | 86.26 | 38.99 | 26.34 |
> | MedCopilot-7B | 21.43 | 3.84 | 88.41 | 41.03 | 27.66 |
>
> Following your suggestion, we will move this human evaluation of trajectory to section 5.1 and emphasize the critical importance of the soundness of the intermediate reasoning trajectories in the main text, rather than focusing exclusively on correctness.
>
> > W2 & Q3: Analysis for OOD drop.
>
> **A**: We appreciate your careful observation of the performance gap between ID and OOD. We would like to highlight that this decline mainly stems from **intrinsic task complexity and distribution shift** rather than overfitting. The external suites were intentionally chosen to stress different code‑centric skills (e.g., sequential SQL, hybrid text–table consistency, raw EHR time series, biostatistical power analysis), which induces markedly different agent trajectories (see Fig. 10(b), illustrating divergence in interaction complexity between ID and OOD tasks) and **naturally yield lower absolute scores across all models, including proprietary baselines**. We further explain this with three key empirical observations:
>
> * **The drop reflects task/data complexity, not memorization (Table 11\)**. We quantify a “difficulty gap” by comparing models on the same task with raw vs. fully processed inputs in Table 11\. On MIMIC‑Extract, gpt‑4.1 falls from 28.94% (processed) to 10.41% (raw), and similar collapses hold for other models. This isolates data complexity (feature engineering on unprocessed time‑series data) as the driver of lower absolute OOD scores rather than a failure to generalize.
> * **Commercial and OSS models exhibit parallel OOD degradation (Tables 3 & 5).** If our model were overfitted, it would suffer disproportionately compared to robust generalist models. Instead, commercial models experience a parallel degradation on these challenging OOD tasks. For example, gpt-4.1 drops from an average of 70.15% on the internal test set to 36.15% on the external OOD suite. The consistency of this trend across model families indicates that the difficulty is intrinsic to the OOD tasks rather than specific to our training process.
> * **Med‑Copilot improves substantially over its base models on OOD tasks (Table 5).** Despite the harder setting, Med‑Copilot‑14B (GRPO) achieves 47.02% on the external suite, a \+19.10% gain over its backbone (Qwen‑2.5‑14B‑Instruct: 27.92%). Similarly, Med‑Copilot‑7B (GRPO) improves from 20.60% → 36.63% over the 7B backbone. This significant gain on unseen distributions verifies that MedAgentGym instills transferable biomedical coding proficiency rather than memorizing training trajectories.
>
> We will add an in-depth analysis of the performance gap between ID and OOD in the updated manuscript.

---

> > ### Author Response · Authors · 2025-11-21
> > **Response to Reviewer 7VJa -- Part II**
> >
> > > Q2: Data Decontamination.
> >
> > **A:** We acknowledge the possibility of data contamination in the pre-training of proprietary LLMs. To mitigate this, we have taken several steps as described in Section 3.2: (1) **Restricted Access and Credentialing**: We constructed many datasets from protected data that cannot be used for proprietary LLMs training (e.g., MIMIC, EHRSHOT, EHR-SeqSQL, EHRCon, MIMIC-Extract) or very recent sources where possible (e.g., MedAgentBench, BioDSBench). (2) **Newly Curated Samples:** For N-PowerAI, we manually curated samples rather than a public repository, effectively creating a private evaluation set. (3) **Rigorous N-Gram Decontamination:** We performed n-gram overlap checks to eliminate direct contamination between our training and test splits. In addition, MedAgentGym includes complex, multi-step reasoning tasks that cannot be solved by simple memorization. We also report on OOD datasets using only their test sets for evaluation (Table 5), further reducing any coupling between in-distribution training and testing. We will add the discussion regarding data contamination in the limitation section.
> >
> > > Q4: Variance over mean.
> >
> > **A**: We appreciate the reviewer’s suggestion regarding statistical rigor. In this work, we focus on
> > large‑scale, execution‑verified single‑run evaluation due to several practical and methodological reasons:
> >
> > * **Computational Intensity of Agentic Benchmarks:** Unlike standard single-turn QA benchmarks, MedAgentGym involves complex, multi-turn agentic interactions (up to 15 turns per instance) involving code generation, execution, and debugging loops. Repeating this process multiple times for every baseline, particularly for API-based models, is prohibitively expensive, aligning with standard practices in large-scale agentic benchmarks (e.g., SWE-bench, WebArena) which typically report single-pass metrics.
> > * **Statistical Stability via Large Sample Size:** Given the large test set (N=4,754 instances), the margin of error is intrinsically low. Furthermore, the performance gaps we report are substantial (e.g., a 19.1% improvement on OOD tasks), ensuring that the reported improvements are statistically significant signals rather than noise.
> > * **Deterministic Reproducibility Protocols:** To minimize stochasticity and ensure reproducibility, we utilized T=0 for all instruction-following baselines (T=0.7 for reasoning baselines) and a fixed scaffold, so repeated evaluations would be nearly deterministic given the same prompts and environment.
> >
> > We will clarify these constraints in the updated manuscript and explicitly state that MedAgentGym focuses on **large‑scale, execution‑verified single-run estimates** under a fixed scaffold, with released code enabling the community to augment our results with more extensive uncertainty analyses where resources permit.
> >
> > ***
> > Thank you again for your review. We hope our response could address your concerns. If you have any further questions, we would be happy to discuss them further.

---

### Author Response · Authors · 2025-11-28

Dear PC, SAC, AC, and Reviewers,

We sincerely thank you for your constructive feedback and expert insights, which have substantially strengthened the manuscript. We are encouraged that all reviewers recognized MedAgentGym as a comprehensive and effective environment for improving coding-based medical reasoning capabilities of LLM agents.

Based on your feedback, we have made the following updates to the paper:

* We added new experimental results for large-scale medical reasoning models in **Table 3** to provide a more equitable and rigorous comparison and validate our previous observations.
* We added new experimental results for a very recent coding-specific model in **Table 3** to augment our previous coding-specific model results and validate our previous observations.
* We added new experimental results on optional reward design in **Appendix F** to guide users who prefer more granular rewards; these are implementation-level options, not prerequisites for the reported gains.
* We expanded **Table 8 in Appendix F** to include additional evaluation of code quality and efficiency across in-distribution and out-of-distribution tasks.
* We added several clarification sentences in the main paper to avoid ambiguity and improve readability.

We provide further details in our point-by-point responses and have incorporated all additions into the revised manuscript. We are pleased that our rebuttal **has successfully addressed the main concerns of reviewers cegQ, xZ94, and Keuw,** and we greatly appreciate all reviewers’ encouragement in raising or finalizing their scores to 8 (conf. 3, reviewer Keuw), 6 (conf. 3, reviewer xZ94), 8 (conf. 3, reviewer cegQ), and 8 (conf. 4, reviewer 7VJa), **leading to a final overall score of 7.5 (before review reverting)**.

For completeness, we note that all updates summarized above, including author-reviewer discussions and score updates, **were completed by Nov 25, well before the widely reported large-scale leak of reviewer/AC identities on Nov 27**. Our revisions were guided solely by the scientific content of the reviews and by the standard rebuttal process.

Thank you again for your time and dedication in reviewing our submission.

Sincerely,

The Authors of Submission \#23682

---

### Meta-Review · Area_Chair_Qzf4 · 2026-01-07

**Summary:**

The last two reviewers confirmed that their major concerns (mainly on the clarity / distinctions over standard coding benchmarks) are addressed. This leads to overall positive ratings across reviewers.

Reviewers have praised multiple merits of the manuscript: the significance of the problem settings, the comprehensive and realism of benchmark, rigorous evaluation, incorporation of open-source models (important for privacy-sensitive biomedical settings), and good software engineering considerations for reproducibility. Based on the further addresses of most raised concerns, the manuscript reaches a high quality and relevance that may be worthy of sharing through oral presentations.

**Reviewer Concerns:**

`7VJa`: lack of assessments on intermediate steps; OOD behavior indicating overfitting; concerns over contaminations; statistical analysis – mostly addressed through additional clarifications and experiments.

`cegQ`: suggestions on the title; expecting user instructions – mostly addressed

 `xZ94`: risk of overfitting, key technical components already exist, expecting comparison with larger models -- main concerns are conformed to be addressed and a score was raised by the reviewer on Nov. 26

`Keuw`: difference from generic coding tasks/benchmakrs and dimensions of assessments; missing details on data construction; more qualitative examples needed; questions on presentations -- main concerns are conformed to be addressed by the reviewer on Nov. 26

**Reviewer Scores:**

Last two reviewers confirmed address of major concerns, leading to overall positive ratings. The high quality and relevance of the manuscript are acknowledged by reviewers.

---

### Decision · Program_Chairs · 2026-01-26

Accept (Oral)